# Autofocused oracles for model-based design

**Clara Fannjiang and Jennifer Listgarten**
Department of Electrical Engineering & Computer Sciences
University of California, Berkeley
Berkeley, CA 94720
{clarafy,jennl}@berkeley.edu

## Abstract

Data-driven design is making headway into a number of application areas, including protein, small-molecule, and materials engineering. The design goal is to construct an object with desired properties, such as a protein that binds to a therapeutic target, or a superconducting material with a higher critical temperature than previously observed. To that end, costly experimental measurements are being replaced with calls to high-capacity regression models trained on labeled data, which can be leveraged in an *in silico* search for design candidates. However, the design goal necessitates moving into regions of the design space beyond where such models were trained. Therefore, one can ask: should the regression model be altered as the design algorithm explores the design space, in the absence of new data? Herein, we answer this question in the affirmative. In particular, we (i) formalize the data-driven design problem as a non-zero-sum game, (ii) develop a principled strategy for retraining the regression model as the design algorithm proceeds—what we refer to as *autofocusing*, and (iii) demonstrate the promise of autofocusing empirically.

## 1 Oracle-based design

The design of objects with desired properties, such as novel proteins, molecules, or materials, has a rich history in bioengineering, chemistry, and materials science. In these domains, design has historically been performed through iterative, labor-intensive experimentation [1] (*e.g.*, measuring protein binding affinity) or compute-intensive physics simulations [2] (*e.g.*, computing low-energy structures for nanomaterials). Increasingly, however, attempts are being made to replace these costly and time-consuming steps with cheap and fast calls to a proxy regression model, trained on labeled data [3, 4, 5, 6, 7]. Herein, we refer to such a proxy model as an *oracle*, and assume that acquisition of training data for the oracle is complete, as in [5, 6, 7, 8, 9].[1] The key issue addressed by our work is how best to train an oracle for use in design, given fixed training data.

In contrast to the traditional use of predictive models, oracle-based design is distinguished by the fact that it seeks solutions—and therefore, will query the oracle—in regions of the design space that are not well-represented by the oracle training data. If this is not the case, the design problem is easy in that the solution is within the region of the training data. Furthermore, one does not know beforehand which parts of the design space a design procedure will navigate through. As such, a major challenge arises when an oracle is employed for design: its outputs, including its uncertainty estimates, become unreliable beyond the training data [10, 11]. Successful oracle-based design thus involves an inherent trade-off between the need to stay "near" the training data in order to trust the oracle, and the need to depart from it in order to make improvements. While trust region approaches have been developed

to help address this trade-off [11, 7], herein, we take a different approach and ask: *what is the most effective way to use a fixed, labeled dataset to train an oracle for design*?

**Contributions** We develop a novel approach to oracle-based design that specifies how to update the oracle as the design space is explored—what we call *autofocusing* the oracle. In particular, we (i) formalize oracle-based design as a non-zero-sum game, (ii) derive an oracle-updating strategy for seeking a Nash equilibrium, and (iii) demonstrate empirically that autofocusing holds promise for improving oracle-based design.

## 2 Model-based optimization for design

Design problems can be cast as seeking points in the design space, $\mathbf{x} \in \mathcal{X}$, that with high probability satisfy desired conditions on a property random variable, $y \in \mathbb{R}$. For example, one might want to design a superconducting material by specifying its chemical composition, $\mathbf{x}$, such that the resulting material has critical temperature greater than some threshold, $y \geq y_\tau$, or has maximal critical temperature, $y = y_{\max}$. We specify the desired properties using a constraint set, $S$, such as $S = \{y \colon y \geq y_\tau\}$ for some $y_\tau$. The design goal is then to solve $\arg\max_{\mathbf{x}} P(y \in S \mid \mathbf{x})$. This optimization problem over the inputs, $\mathbf{x}$, can be converted to one over *distributions* over the design space [12, 13]. Specifically, model-based optimization (MBO) seeks the parameters, $\theta$, of a "search model", $p_\theta(\mathbf{x})$, that maximizes an objective that bounds the original objective:

$$\max_{\mathbf{x}} P(y \in S \mid \mathbf{x}) \geq \max_{\theta \in \Theta} \mathbb{E}_{p_\theta(\mathbf{x})}[P(y \in S \mid \mathbf{x})] = \max_{\theta \in \Theta} \mathbb{E}_{p_\theta(\mathbf{x})}\left[\int_S p(y \mid \mathbf{x}) dy\right]. \quad (1)$$

The original optimization problem over $\mathbf{x}$, and the MBO problem over $\theta$, are equivalent when the search model has the capacity to place point masses on optima of the original objective. Reasons for using the MBO formulation include that it requires no gradients of $p(y \mid \mathbf{x})$, thereby allowing the use of arbitrary oracles for design, including those that are not differentiable with respect to the design space and otherwise require specialized treatment [14]. MBO also naturally allows one to obtain not just a single design candidate, but a diverse set of candidates, by sampling from the final search distribution (whose entropy can be adjusted by adding regularization to Equation 1). Finally, MBO introduces the language of probability into the optimization, thereby allowing coherent incorporation of probabilistic constraints such as implicit trust regions [11]. The search model can be any parameterized probability distribution that can be sampled from, and whose parameters can be estimated using weighted maximum likelihood estimation (MLE) or approximations thereof. Examples include mixtures of Gaussians, hidden Markov models, variational autoencoders [15], and Potts models [16]. Notably, the search model distribution can be over discrete or continuous random variables, or a combination thereof.

We use the phrase *model-based design* (MBD) to denote use of MBO to solve a design problem. Hereafter, we focus on oracle-based MBD, which attempts to solve Equation 1 by replacing costly and time-consuming queries of the ground truth[2], $p(y \mid \mathbf{x})$, with calls to a trained regression model (*i.e.*, oracle), $p_\beta(y \mid \mathbf{x})$, with parameters, $\beta \in B$. Given access to a fixed dataset, $\{(\mathbf{x}_i, y_i)\}_{i=1}^n$, the oracle is typically trained once using standard techniques and thereafter considered fixed [5, 6, 7, 8, 9, 11, 17, 18]. In what follows, we describe why such a strategy is sub-optimal and how to re-train the oracle in order to better achieve design goals. First, however, we briefly review a common approach for performing MBO, as we will leverage such algorithms in our approach.

### 2.1 Solving model-based optimization problems

MBO problems are often tackled with an Estimation of Distribution Algorithm (EDA) [19, 20], a class of iterative optimization algorithms that can be seen as Monte Carlo expectation-maximization [13]; EDAs are also connected to the cross-entropy method [21, 22] and reward-weighted regression in

reinforcement learning [23]. Given an oracle, $p_\beta(y \mid \mathbf{x})$, and an initial search model, $p_{\theta^{(t=0)}}$, an EDA typically proceeds at iteration $t$ with two core steps:

1. "E-step": Sample from the current search model, $\tilde{\mathbf{x}}_i \sim p_{\theta^{(t-1)}}(\mathbf{x})$ for all $i \in \{1, \dots, m\}$. Compute a weight for each sample, $v_i := V(P_\beta(y \in S \mid \tilde{\mathbf{x}}_i))$, where $V(.)$ is a method-specific, monotonic transformation.

2. "M-step": Perform weighted MLE to yield an updated search model, $p_{\theta^{(t)}}(\mathbf{x})$, which tends to have more mass where $P_\beta(y \in S \mid \mathbf{x})$ is high. (Some EDAs can be seen as performing *maximum a posteriori* inference instead, which results in smoothed parameter updates [11].)

Upon convergence of the EDA, design candidates can be sampled from the final search model if it is not a point mass; one may also choose to use promising samples from earlier iterations. Notably, the oracle, $p_\beta(y \mid \mathbf{x})$, remains fixed in the steps above. Next, we motivate a new formalism for oracle-based MBD that yields a principled approach for updating the oracle at each iteration.

## 3 Autofocused oracles for model-based design

The common approach of substituting the oracle, $p_\beta(y \mid \mathbf{x})$, for the ground-truth, $p(y \mid \mathbf{x})$, does not address the fact that the oracle is only likely to be reliable over the distribution from which its training data were drawn [10, 24, 25]. To address this problem, we now reformulate the MBD problem as a non-zero-sum game, which suggests an algorithmic strategy for iteratively updating the oracle within any MBO algorithm.

### 3.1 Model-based design as a game

When the objective in Equation 1 is replaced with an oracle-based version,

$$\arg\max_{\theta \in \Theta} \mathbb{E}_{p_\theta(\mathbf{x})}[P_\beta(y \in S \mid \mathbf{x})], \tag{2}$$

the solution to the oracle-based problem will, in general, be sub-optimal with respect to the original objective that uses the ground truth, $P(y \in S \mid \mathbf{x})$. This sub-optimality can be extreme due to pathological behavior of the oracle when the search model, $p_\theta(\mathbf{x})$, strays too far from the training distribution during the optimization [11].

Since one cannot access the ground truth, we seek a practical alternative wherein we can leverage an oracle, but also infer when the values of the ground-truth and oracle-based objectives (in Equations 1 and 2, respectively) are likely to be close. To do so, we introduce the notion of the *oracle gap*, defined as $\mathbb{E}_{p_\theta(\mathbf{x})}[|P(y \in S \mid \mathbf{x}) - P_\beta(y \in S \mid \mathbf{x})|]$. When this quantity is small, then by Jensen's inequality the oracle-based and ground-truth objectives are close. Consequently, our insight for improving oracle-based design is to use the oracle that minimizes the oracle gap,

$$\arg\min_{\beta \in B} \text{ORACLEGAP}(\theta, \beta) = \arg\min_{\beta \in B} \mathbb{E}_{p_\theta(\mathbf{x})}[|P(y \in S \mid \mathbf{x}) - P_\beta(y \in S \mid \mathbf{x})|]. \tag{3}$$

Together, Equations 2 and 3 define the coupled objectives of two players, namely the search model (with parameters $\theta$) and the oracle (with parameters $\beta$), in a non-zero-sum game. To attain good objective values for both players, our goal will be to search for a Nash equilibrium—that is, a pair of values $(\theta^*, \beta^*)$ such that neither can improve its objective given the other. To do so, we develop an alternating ascent-descent algorithm, which alternates between (i) fixing the oracle parameters and updating the search model parameters to increase the objective in Equation 2 (the ascent step), and (ii) fixing the search model parameters and updating the oracle parameters to decrease the objective in Equation 3 (the descent step). In the next section, we describe this algorithm in more detail.

**Practical interpretation of the MBD game.**   Interpreting the usefulness of this game formulation requires some subtlety. The claim is not that every Nash equilibrium yields a search model that provides a high value of the (unknowable) ground-truth objective in Equation 1. However, for any pair of values, $(\theta, \beta)$, the value of the oracle gap provides a certificate on the value of the ground-truth objective. In particular, if one has an oracle and search model that yield an oracle gap of $\epsilon$, then by Jensen's inequality the ground-truth objective is within $\epsilon$ of the oracle-based objective. Therefore, to the extent that we are able to minimize the oracle gap (Equation 3), we can trust the value of our

oracle-based objective (Equation 2). Note that a small, or even zero oracle gap only implies that the oracle-based objective is trustworthy; successful design also entails achieving a *high* oracle-based objective, the potential for which depends on an appropriate oracle class and suitably informative training data (as it always does for oracle-based design, regardless of whether our framework is used).

Although the oracle gap as a certificate is useful conceptually for motivating our approach, at present it is not clear how to estimate it. In our experiments, we found that we could demonstrate the benefits of autofocusing without directly estimating the oracle gap, relying solely on the principle of minimizing it. We also note that in practice, what matters is not whether we converge to a Nash equilibrium, just as what matters in empirical risk minimization is not whether one exactly recovers the global optimum, only a useful point. That is, if we can find parameters, $(\theta, \beta)$, that yield better designs than alternative methods, then we have developed a useful method.

## 3.2 An alternating ascent-descent algorithm for the MBD game

Our approach alternates between an ascent step that updates the search model, and a descent step that updates the oracle. The ascent step is relatively straightforward as it leverages existing MBO algorithms. The descent step, however, requires some creativity. In particular, for the ascent step, we run a single iteration of an MBO algorithm as described in §2.1, to obtain a search model that increases the objective in Equation 2. For the descent step, we aim to minimize the oracle gap in Equation 3 by making use of the following observation (proof in Supplementary Material §S2).

**Proposition 1.** *For any search model, $p_\theta(\mathbf{x})$, if the oracle parameters, $\beta$, satisfy*

$$\mathbb{E}_{p_\theta(\mathbf{x})}[D_{KL}(p(y \mid \mathbf{x}) \mid\mid p_\beta(y \mid \mathbf{x}))] = \int_{\mathcal{X}} D_{KL}(p(y \mid \mathbf{x}) \mid\mid p_\beta(y \mid \mathbf{x})) \, p_\theta(\mathbf{x}) d\mathbf{x} \leq \epsilon, \qquad (4)$$

*where $D_{KL}(p \mid\mid q)$ is the Kullback-Leibler (KL) divergence between distributions $p$ and $q$, then the following bound holds:*

$$\mathbb{E}_{p_\theta(\mathbf{x})}[|P(y \in S \mid \mathbf{x}) - P_\beta(y \in S \mid \mathbf{x})|] \leq \sqrt{\frac{\epsilon}{2}}. \qquad (5)$$

As a consequence of Proposition 1, given any search model, $p_\theta(\mathbf{x})$, an oracle that minimizes the expected KL divergence in Equation 4 also minimizes an upper bound on the oracle gap. Our descent strategy is therefore to minimize this expected divergence. In particular, as shown in the Supplementary Material §S2, the resulting oracle parameter update at iteration $t$ can be written as $\beta^{(t)} = \arg\max_{\beta \in B} \mathbb{E}_{p_{\theta^{(t)}}(\mathbf{x})} \mathbb{E}_{p(y|\mathbf{x})}[\log p_\beta(y \mid \mathbf{x})]$, where we refer to the objective as the log-likelihood under the search model. Although we cannot generally access the ground truth, $p(y \mid \mathbf{x})$, we do have labeled training data, $\{(\mathbf{x}_i, y_i)\}_{i=1}^n$, whose labels come from the ground-truth distribution, $y_i \sim p(y \mid \mathbf{x} = \mathbf{x}_i)$. We therefore use importance sampling with the training distribution, $p_0(\mathbf{x})$, as the proposal distribution, to obtain a now-practical oracle parameter update,

$$\beta^{(t)} = \arg\max_{\beta \in B} \frac{1}{n} \sum_{i=1}^n \frac{p_{\theta^{(t)}}(\mathbf{x}_i)}{p_0(\mathbf{x}_i)} \log p_\beta(y_i \mid \mathbf{x}_i). \qquad (6)$$

The training points, $\mathbf{x}_i$, are used to estimate some model for $p_0(\mathbf{x})$, while $p_{\theta^{(t)}}(\mathbf{x})$ is given by the search model. We discuss the variance of the importance weights, $w_i := p_\theta(\mathbf{x}_i)/p_0(\mathbf{x}_i)$, shortly.

Together, the ascent and descent steps amount to appending a "Step 3" to each iteration of the generic two-step MBO algorithm outlined in §2.1, in which the oracle is retrained on re-weighted training data according to Equation 6. We call this strategy *autofocusing* the oracle, as it retrains the oracle in lockstep with the search model, to keep the oracle likelihood maximized on the most promising regions of the design space. Pseudo-code for autofocusing can be found in the Supplementary Material (Algorithms 1 and 2). As shown in the experiments, autofocusing tends to improve the outcomes of design procedures, and when it does not, no harm is incurred relative to the naive approach with a fixed oracle. Before discussing such experiments, we first make some remarks.

## 3.3 Remarks on autofocusing

**Controlling variance of the importance weights.** It is well known that importance weights can have high, even infinite, variance [26], which may prevent the importance-sampled estimate of the

log-likelihood from being useful for retraining the oracle effectively. That is, solving Equation 6 may not reliably yield oracle parameter estimates that minimize the log-likelihood under the search model. To monitor the reliability of the importance-sampled estimate, one can compute and track an *effective sample size* of the re-weighted training data, $n_e := (\sum_{i=1}^n w_i)^2 / \sum_{i=1}^n w_i^2$, which reflects the variance of the importance weights [26]. If one has some sense of a suitable sample size for the application at hand (*e.g.*, based on the oracle model capacity), then one could monitor $n_e$ and choose not to retrain when it is too small. Another variance control strategy is to use a trust region to constrain the movement of the search model, such as in [11], which automatically controls the variance (see Supplementary Material Proposition S2.2). Indeed, our experiments show how autofocusing works synergistically with a trust-region approach. Finally, two other common strategies are: (i) self-normalizing the weights, which provides a biased but consistent and lower-variance estimate [26], and (ii) flattening the weights [24] to $w_i^\alpha$ according to a hyperparameter, $\alpha \in [0, 1]$. The value of $\alpha$ interpolates between the original importance weights ($\alpha = 1$), which provide an unbiased but high-variance estimate, and all weights equal to one ($\alpha = 0$), which is equivalent to naively training the oracle (*i.e.*, no autofocusing).

**Oracle bias-variance trade-off.** If the oracle equals the ground truth over all parts of the design space encountered during the design procedure, then autofocusing should not improve upon using a fixed oracle. In practice, however, this is unlikely to ever be the case—the oracle is almost certain to be misspecified and ultimately mislead the design procedure with incorrect inductive bias. It is therefore interesting to consider what autofocusing does from the perspective of the bias-variance trade-off of the oracle, with respect to the search model distribution. On the one hand, autofocusing retrains the oracle using an unbiased estimate of the log-likelihood over the search model. On the other hand, as the search model moves further away from the training data, the effective sample size available to train the oracle decreases; correspondingly, the variance of the oracle increases. In other words, when we use a fixed oracle (no autofocusing), we prioritize minimal variance at the expense of greater bias. With pure autofocusing, we prioritize reduction in bias at the expense of higher variance. Autofocusing with techniques to control the variance of the importance weights [24, 27] enables us to make a suitable trade-off between these two extremes.

**Autofocusing corrects design-induced covariate shift.** In adopting an importance-sampled estimate of the training objective, Equation 6 is analogous to the classic covariate shift adaptation strategy known as importance-weighted empirical risk minimization [24, 27]. We can therefore interpret autofocusing as dynamically correcting for covariate shift induced by a design procedure, where, at each iteration, a new "test" distribution is given by the updated search model. Furthermore, we are in the fortunate position of knowing the exact parametric form of the test density at each iteration, which is simply that of the search model. This view highlights that the goal of autofocusing is not necessarily to increase exploration of the design space, but to provide a more useful oracle wherever the search model does move (as dictated by the underlying method to which autofocusing is added).

## 4 Related Work

Although there is no cohesive literature on oracle-based design in the fixed-data setting, its use is gaining prominence in several application areas, including the design of proteins and nucleotide sequences [7, 11, 17, 18, 28], molecules [29, 8, 9], and materials [6, 30]. Within such work, the danger in extrapolating beyond the training distribution is not always acknowledged or addressed. In fact, proposed design procedures often are validated under the assumption that the oracle is always correct [8, 14, 17, 18, 29]. Some exceptions include Conditioning by Adaptive Sampling (CbAS) [11], which employs a probabilistic trust-region approach using a model of the training distribution, and [7], which uses a hard distance-based threshold. Similar in spirit to [11], Linder et al. regularize the designed sequences based on their likelihood under a model of the training distribution [31]. In another approach, a variational autoencoder implicitly enforces a trust region by constraining design candidates to the probabilistic image of the decoder [9]. Finally, Kumar & Levine tackle design by learning the inverse of a ground-truth function, which they constrain to agree with an oracle, so as to discourage too much departure from the training data [28]. None of these approaches update the oracle in any way. However, autofocusing is entirely complementary to and does not preclude the additional use of any of these approaches. For example, we demonstrate in our experiments that autofocusing improves the outcomes of CbAS, which implicitly inhibits the movement of the search model away from the training distribution.

Related to the design problem is that of active learning in order to optimize a function, using for example Bayesian optimization [32]. Such approaches are fundamentally distinct from our setting in that they dynamically acquire new labeled data, thereby more readily allowing for correction of oracle modeling errors. In a similar spirit, evolutionary algorithms sometimes use a "surrogate" model of the function of interest (equivalent to our oracle), to help guide the acquisition of new data [33]. In such settings, the surrogate may be updated using an *ad hoc* subset of the data [34] or perturbation of the surrogate parameters [35]. Similarly, a recent reinforcement-learning based approach to biological sequence design relies on new data to refine the oracle when moving into a region of design space where the oracle is unreliable [36].

Offline reinforcement learning (RL) [37] shares similar characteristics with our problem in that the goal is to find a policy that optimizes a reward function, given only a fixed dataset of trajectories sampled using another policy. In particular, offline model-based RL leverages a learned model of dynamics that may not be accurate everywhere. Methods in that setting have attempted to account for the shift away from the training distribution using uncertainty estimation and trust-region approaches [38, 39, 40]; importance sampling has also been used for off-policy evaluation [41, 42].

As noted in the previous section, autofocusing operates through iterative retraining of the oracle in order to correct for covariate shift induced by the movement of the search model. It can therefore be connected to ideas from domain adaptation more broadly [25]. Finally, we note that mathematically, oracle-based MBD is related to the decision-theoretic framework of performative prediction [43]. Perdomo *et al.* formalize the phenomenon in which using predictive models to perform actions induces distributional shift, then present theoretical analysis of repeated retraining with new data as a solution. Our problem has two major distinctions from this setting: first, the ultimate goal in design is to maximize an unknowable ground-truth objective, not to minimize risk of the oracle. The latter is only relevant to the extent that it helps us achieve the former, and our work operationalizes that connection by formulating and minimizing the oracle gap. Second, we are in a fixed-data setting. Our work demonstrates the utility of adaptive retraining even in the absence of new data.

## 5 Experiments

We now demonstrate empirically, across a variety of both experimental settings and MBO algorithms, how autofocusing can help us better achieve design goals. First we leverage an intuitive example to gain detailed insights into how autofocus behaves. We then conduct a detailed study on a more realistic problem of designing superconducting materials. Code for our experiments is available at https://github.com/clarafy/autofocused_oracles.

### 5.1 An illustrative example

To investigate how autofocusing works in a setting that can be understood intuitively, we constructed a one-dimensional design problem where the goal was to maximize a multi-modal ground-truth function, $f(\mathbf{x}) : \mathbb{R} \to \mathbb{R}^+$, given fixed training data (Figure 1a). The training distribution from which training points were drawn, $p_0(\mathbf{x})$, was a Gaussian with variance, $\sigma_0^2$, centered at 3, a point where $f(\mathbf{x})$ is small relative to the global maximum at 7. This captures the common scenario where the oracle training data do not extend out to global optima of the property of interest. As we increase the variance of the training distribution, $\sigma_0^2$, the training data become more and more likely to approach the global maximum of $f(\mathbf{x})$. The training labels are drawn from $p(y \mid \mathbf{x}) = \mathcal{N}(f(\mathbf{x}), \sigma_\epsilon^2)$, where $\sigma_\epsilon^2$ is the variance of the label noise. For this example, we used CbAS [11], an MBO algorithm that employs a probabilistic trust region. We did not control the variance of the importance weights.

An MBO algorithm prescribes a sequence of search models as the optimization proceeds, where each successive search model is fit using weighted MLE to samples from its predecessor. However, in our one-dimensional example, one can instead use numerical quadrature to directly compute each successive search model [11]. Such an approach enables us to abstract out the particular parametric form of the search model, thereby more directly exposing the effects of autofocusing. In particular, we used numerical quadrature to compute the search model density at iteration $t$ as $p^{(t)}(\mathbf{x}) \propto P_{\beta^{(t)}}(y \in S^{(t)} \mid \mathbf{x}) p_0(\mathbf{x})$, where $S^{(t)}$ belongs to a sequence of relaxed constraint sets such that $S^{(t)} \supseteq S^{(t+1)} \supseteq S$ [11]. We computed this sequence of search models in two ways: (i) without autofocusing, that is, with a fixed oracle trained once on equally weighted training data, and (ii) with autofocusing, that is, where the oracle was retrained at each iteration. In both cases, the oracle

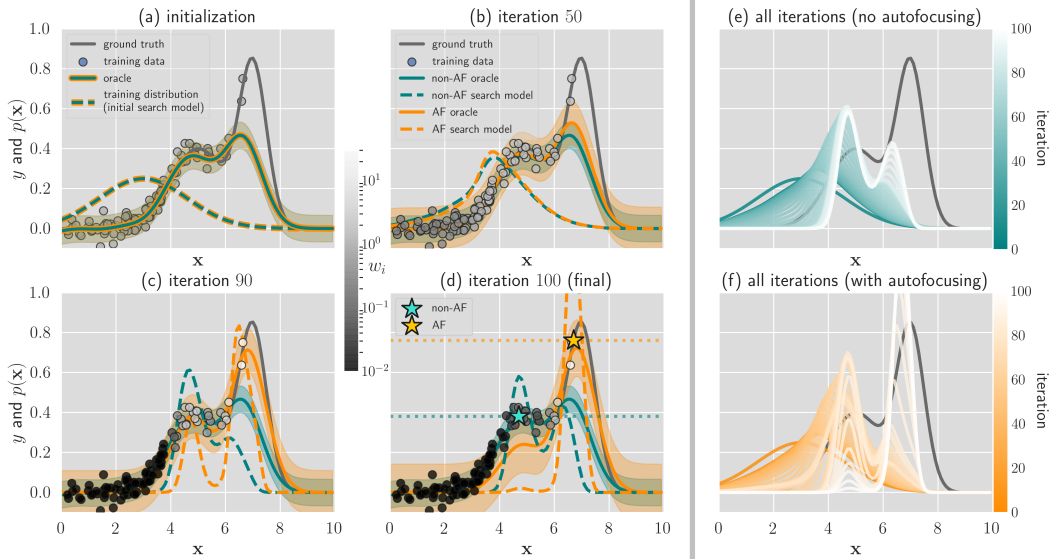

Figure 1: Illustrative example. Panels (a-d) show detailed snapshots of the MBO algorithm, CbAS [11], with and without autofocusing (AF) in each panel. The vertical axis represents both $y$ values (for the oracle and ground truth) and probability density values (of the training distribution, $p_0(\mathbf{x})$, and search distributions, $p_{\theta^{(t)}}(\mathbf{x})$). Shaded envelopes correspond to $\pm 1$ standard deviation of the oracles, $\sigma_{\beta^{(t)}}$, with the oracle expectations, $\mu_{\beta^{(t)}}(\mathbf{x})$, shown as a solid line. Specifically, (a) at initialization, the oracle and search model are the same for AF and non-AF. Intermediate and final iterations are shown in (b-d), where the non-AF and AF oracles and search models increasingly diverge. Greyscale of training points corresponds to their importance weights used for autofocusing. In (d), each star and dotted horizontal line indicate the ground-truth value corresponding to the point of maximum density, indicative of the quality of the final search model (higher is better). The values of $(\sigma_\epsilon, \sigma_0)$ used here correspond to the ones marked by an $\times$ in Figure 2, which summarizes results across a range of settings. Panels (e,f) show the search model over all iterations without and with autofocusing, respectively.

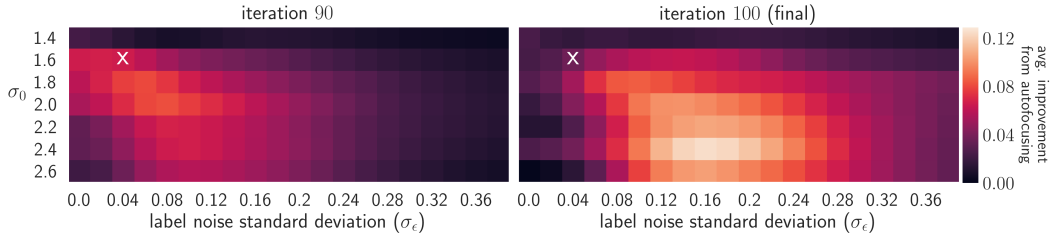

Figure 2: Improvement from autofocusing (AF) over a wide range of settings of the illustrative example. Each colored square shows the improvement (averaged over 50 trials) conferred by AF for one setting, $(\sigma_\epsilon, \sigma_0)$, of, respectively, the standard deviations of the training distribution and the label noise. Improvement is quantified as the difference between the ground-truth objective in Equation 1 achieved by the final search model with and without AF. A positive value means AF yielded higher ground-truth values (*i.e.*, performed better than without AF), while zero means it neither helped nor hurt. Similar plots to Figure 1 are shown in the Supplementary Material for other settings (Figure S1).

was of the form $p_\beta(y \mid \mathbf{x}) = \mathcal{N}(\mu_\beta(\mathbf{x}), \sigma_\beta^2)$, where $\mu_\beta(\mathbf{x})$ was fit by kernel ridge regression with a radial basis function kernel and $\sigma_\beta^2$ was set to the mean squared error between $\mu_\beta(\mathbf{x})$ and the labels (see Supplementary Material §S3 for more details). Since this was a maximization problem, the desired condition was set as $S = \{y : y \geq \max_{\mathbf{x}} \mu_\beta(\mathbf{x})\}$ (where $\mu_\beta(\mathbf{x}) = 0.68$ for the initial oracle). We found that autofocusing more effectively shifts the search model toward the ground-truth global maximum as the iterations proceed (Figure 1b-f), thereby providing improved design candidates.

To understand the effect of autofocusing more systematically, we repeated the experiment just described across a range of settings of the variances of the training distribution, $\sigma_0^2$, and of the label

Table 1: Designing superconducting materials. We ran six different MBO methods, each with and without autofocusing. For each method, we extracted those samples with oracle expectations above the $80^{th}$ percentile and computed their ground-truth expectations. We report the median and maximum of those ground-truth expectations (both in degrees K), their percent chance of improvement (PCI, in percent) over the maximum label in the training data, as well as the Spearman correlation ($\rho$) and root mean squared error (RMSE, in degrees K) between the oracle and ground-truth expectations. Each reported score is averaged over 10 trials, where, in each trial, a different training set was sampled from the training distribution. "Mean Diff." is the average difference between the score when using autofocusing compared to not. Bold values with one star (*) and two stars (**), respectively, mean $p$-values $< 0.05$ and $< 0.01$ from a two-sided Wilcoxon signed-rank test on the 10 paired score differences between a method with autofocus and without ('Original'). For all scores but RMSE, a higher value means autofocusing yielded better results (as indicated by the arrow $\uparrow$); for RMSE, the opposite is true (as indicated by the arrow $\downarrow$).

| | Median $\uparrow$ | Max $\uparrow$ | PCI $\uparrow$ | $\rho \uparrow$ | RMSE $\downarrow$ | Median $\uparrow$ | Max $\uparrow$ | PCI $\uparrow$ | $\rho \uparrow$ | RMSE $\downarrow$ |
|---|---|---|---|---|---|---|---|---|---|---|
| | | | **CbAS** | | | | | **DbAS** | | |
| Original | 51.5 | 103.8 | 0.11 | 0.05 | 17.2 | 57.0 | 98.4 | 0.11 | 0.01 | 29.6 |
| Autofocused | 76.4 | 119.8 | 3.78 | 0.56 | 12.9 | 78.9 | 111.6 | 4.4 | 0.01 | 24.5 |
| Mean Diff. | **24.9**** | **16.0**** | **3.67**** | **0.51**** | **-4.4**** | **21.9**** | **13.2**** | **4.2**** | 0.01 | **-5.1*** |
| | | | **RWR** | | | | | **FB** | | |
| Original | 43.4 | 102.0 | 0.05 | 0.92 | 7.4 | 49.2 | 100.6 | 0.14 | 0.09 | 17.5 |
| Autofocused | 71.4 | 114.0 | 1.60 | 0.65 | 12.7 | 64.2 | 111.6 | 0.86 | 0.49 | 11.1 |
| Mean Diff. | **28.0**** | **12.0**** | **1.56**** | **-0.27**** | **5.4**** | **15.0**** | **11.0**** | **0.73**** | **0.40**** | **-6.4**** |
| | | | **CEM-PI** | | | | | **CMA-ES** | | |
| Original | 34.5 | 55.8 | 0.00 | -0.16 | 148.3 | 42.1 | 69.4 | 0.00 | 0.27 | 27493.2 |
| Autofocused | 67.0 | 98.0 | 1.69 | 0.13 | 29.4 | 50.2 | 85.8 | 0.01 | 0.52 | 9499.8 |
| Mean Diff. | **32.6**** | **42.3*** | **1.69*** | 0.29 | **-118.9**** | **8.1*** | **16.3*** | 0.01 | **0.25*** | **-17993.5*** |

noise, $\sigma_\epsilon^2$ (Figure 2). Intuitively, both these variances control how informative the training data are about the ground-truth global maximum: as $\sigma_0^2$ increases, the training data are more likely to include points near the global maximum, and as $\sigma_\epsilon^2$ decreases, the training labels are less noisy. Therefore, if the training data are either too uninformative (small $\sigma_0^2$ and/or large $\sigma_\epsilon^2$) or too informative (large $\sigma_0^2$ and/or small $\sigma_\epsilon^2$), then one would not expect autofocusing to substantially improve design. In intermediate regimes, autofocusing should be particularly useful. Such a phenomenon is seen in our experiments (Figure 2). Importantly, this kind of intermediate regime is one in which practitioners are likely to find themselves: the motivation for design is often sparked by the existence of a few examples with property values that are exceptional compared to most known examples, yet the design goal is to push the desired property to be more exceptional still. In contrast, if the true global optimum already resides in the training data, one cannot hope to design anything better anyway. However, even in regimes where autofocusing does not help, on average it does not hurt relative to a naive approach with a fixed oracle (Figure 2 and Supplementary Material §5.1).

## 5.2 Designing superconductors with maximal critical temperature

Designing superconducting materials with high critical temperatures is an active research problem that impacts engineering applications from magnetic resonance imaging systems to the Large Hadron Collider. To assess autofocusing in a more realistic scenario, we used a dataset comprising $21,263$ superconducting materials paired with their critical temperatures [44], the maximum temperature at which a material exhibits superconductivity. Each material is represented by a feature vector of length eighty-one, which contains real-valued properties of the material's constituent elements (*e.g.*, their atomic radius and valence). We outline our experiments here, with details deferred to the Supplementary Material §S4.

Unlike *in silico* validation of a predictive model, one cannot hold out data to validate a design algorithm because one will not have ground-truth labels for proposed design candidates. Thus, similarly to [11], we created a "ground-truth" model by training gradient-boosted regression trees [44, 45] on the whole dataset and treating the output as the ground-truth expectation, $\mathbb{E}[y \mid \mathbf{x}]$, which can be called at any time. Next, we generated training data to emulate the common scenario in which design practitioners have labeled data that are not dense near ground-truth global optima. In particular, we selected the $n = 17,015$ training points from the dataset whose ground-truth expectations were in the bottom $80^{th}$ percentile. We used MLE with these points to fit a full-rank multivariate normal,

which served as the training distribution, $p_0(\mathbf{x})$, from which we drew $n$ simulated training points, $\{\mathbf{x}_i\}_{i=1}^n$. For each $\mathbf{x}_i$ we drew one sample, $y_i \sim \mathcal{N}(\mathbb{E}[y \mid \mathbf{x}_i], 1)$, to obtain a noisy ground-truth label. Finally, for our oracle, we used $\{(\mathbf{x}_i, y_i)\}_{i=1}^n$ to train an ensemble of three neural networks that output both $\mu_\beta(\mathbf{x})$ and $\sigma_\beta^2(\mathbf{x})$, to provide predictions of the form $p_\beta(y \mid \mathbf{x}) = \mathcal{N}(\mu_\beta(\mathbf{x}), \sigma_\beta^2(\mathbf{x}))$ [46].

We ran six different MBO algorithms, each with and without autofocusing, with the goal of designing materials with maximal critical temperatures. In all cases, we used a full-rank multivariate normal for the search model, and flattened the importance weights used for autofocusing to $w_i^\alpha$ [24] with $\alpha = 0.2$ to help control variance. The MBO algorithms were: (i) Conditioning by Adaptive Sampling (CbAS) [11]; (ii) Design by Adaptive Sampling (DbAS) [11]; (iii) reward-weighted regression (RWR) [23]; (iv) the "feedback" mechanism proposed in [17] (FB); (v) the cross-entropy method used to optimize probability of improvement (CEM-PI) [11, 32]; and (vi) Covariance Matrix Adaptation Evolution Strategy (CMA-ES) [47]. These are briefly described in the Supplementary Material §S4.

To quantify the success of each algorithm, we did the following. At each iteration, $t$, we first computed the oracle expectations, $\mathbb{E}_{\beta^{(t)}}[y \mid \mathbf{x}]$, for each of $n$ samples drawn from the search model, $p_{\theta^{(t)}}(\mathbf{x})$. We then selected the iteration where the $80^{th}$ percentile of these oracle expectations was greatest. For that iteration, we computed various summary statistics on the *ground-truth* expectations of the best samples, as judged by the oracle from that iteration (*i.e.*, samples with oracle expectations greater than the $80^{th}$ percentile; Table 1). See Algorithm 3 in the Supplementary Material for pseudocode of this procedure. Our evaluation procedure emulates the typical setting in which a practitioner has limited experimental resources, and can only evaluate the ground truth for the most promising candidates [4, 5, 6, 7].

Across the majority of evaluation metrics, for all MBO methods, autofocusing a method provided a statistically significant improvement upon the original method. The percent chances of improvement (PCI, the percent chances that the best samples had greater ground-truth expectations than the maximum label in the training data), expose the challenging nature of the design problem. All methods with no autofocusing had a PCI less than $0.14\%$, which although small, still reflects a marked improvement over a naive baseline of simply drawing $n$ new samples from the training distribution itself, which achieves $5.9 \times 10^{-3}\%$. Plots of design trajectories from these experiments, and results from experiments without variance control and with oracle architectures of higher and lower capacities, can be found in the Supplementary Material (Figures S3 and S4, Table S2).

## 6   Discussion

We have introduced a new formulation of oracle-based design as a non-zero-sum game. From this formulation, we developed a new approach for design wherein the oracle—the predictive model that replaces costly and time-consuming laboratory experiments—is iteratively retrained so as to "autofocus" it on the current region of design candidates under consideration. Our autofocusing approach can be applied to any design procedure that uses model-based optimization. We recommend using autofocusing with an MBO method that uses trust regions, such as CbAS [11], which automatically helps control the variance of the importance weights used for autofocusing. For autofocusing an MBO algorithm without a trust region, practical use of the oracle gap certificate and/or effective sample size should be further investigated. Nevertheless, even without these, we have demonstrated empirically that autofocusing can provide benefits.

Autofocusing can be seen as dynamically correcting for covariate shift as the design procedure explores design space. It can also be understood as enabling a design procedure to navigate a trade-off between the bias and variance of the oracle, with respect to the search model distribution. One extension of this idea is to also perform oracle model selection at each iteration, such that the model capacity is tailored to the level of importance weight variance.

Further extensions to consider are alternate strategies for estimating the importance weights [27]. In particular, training discriminative classifiers to estimate these weights may be fruitful when using search models that are implicit generative models, or whose likelihood cannot otherwise be computed in closed form, such as variational autoencoders [27, 48]. We believe this may be a promising approach for applying autofocusing to biological sequence design and other discrete design problems, which often leverage such models. One can also imagine extensions of autofocusing to gradient-based design procedures [18]—for example, using techniques for test-time oracle retraining, in order to evaluate the current point most accurately [49].

## Acknowledgments and Disclosure of Funding

Many thanks to Sebastián Prillo, David Brookes, Chloe Hsu, Hunter Nisonoff, Akosua Busia, and Sergey Levine for helpful comments on the work. We are also grateful to the U.S. National Science Foundation Graduate Research Fellowship Program and the Chan Zuckerberg Investigator Program for funding.

## 7 Broader Impact

If adopted more broadly, our work could affect how novel proteins, small molecules, materials, and other entities are engineered. Because predictive models are imperfect, even with the advances presented herein, care should be taken by practitioners to verify that any proposed design candidates are indeed safe and ethical for the intended downstream applications. The machine learning approach we present facilitates obtaining promising design candidates in a cost-effective manner, but practitioners must follow up on candidates proposed by our approach with conventional laboratory methods, as appropriate to the application domain.

## Footnotes

[1]For many applications in protein, molecule, and material design, even if one performs iterative rounds of data acquisition, at some point, the acquisition phase concludes due to finite resources.

[2]We refer to the *ground truth* as the distribution of direct property measurements, which are inevitably stochastic due to sensor noise.

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
