[Supplementary Material]

# Supplementary Material

## S1   Pseudocode

Algorithm 1 gives pseudocode for autofocusing a broad class of model-based optimization (MBO) algorithms known as estimation of distribution algorithms (EDAs), which can be seen as performing Monte-Carlo expectation-maximization [13]. EDAs proceed at each iteration with a sampling-based "E-step" (Steps 1 and 2 in Algorithm 1) and a weighted maximum likelihood estimation (MLE) "M-step" (Step 3; see [13] for more details). Different EDAs are distinguished by method-specific monotonic transformations $V(\cdot)$, which determine the sample weights used in the M-step. In some EDAs, this transformation is not explicitly defined, but rather implicitly implemented by constructing and using a sequence of relaxed constraint sets, $S^{(t)}$, such that $S^{(t)} \supseteq S^{(t+1)} \supseteq S$ [21, 22, 11].

Algorithm 2 gives pseudocode for autofocusing a particular EDA, Conditioning by Adaptive Sampling (CbAS) [11], which uses such a sequence of relaxed constraint sets, as well as M-step weights that induce an implicit trust region for the search model update. For simplicity, the algorithm is instantiated with the design goal of maximizing the property of interest. It can easily be generalized to the goal of achieving a specific value for the property, or handling multiple properties (see Sections S2-3 of [11]).

Use of [.] in the pseudocode denotes an optional input argument with default values.

---

**Algorithm 1: Autofocused model-based optimization algorithm**

**Input**       : Training data, $\{(\mathbf{x}_i, y_i)\}_{i=1}^n$; oracle model class, $p_\beta(y \mid \mathbf{x})$ with parameters, $\beta$, that can be estimated with MLE; search model class, $p_\theta(\mathbf{x})$, with parameters, $\theta$, that can be estimated with weighted MLE or approximations thereof; desired constraint set, $S$ (*e.g.*, $S = \{y \mid y \geq y_\tau.\}$); maximum number of iterations, $T$; number of samples to generate, $m$; EDA-specific monotonic transformation, $V(\cdot)$.

**Initialization :** Obtain $p_0(\mathbf{x})$ by fitting to $\{\mathbf{x}_i\}_{i=1}^n$ with the search model class. For the search model, set $p_{\theta^{(0)}}(\mathbf{x}) \leftarrow p_0(\mathbf{x})$. For the oracle, $p_{\beta^{(0)}}(y \mid \mathbf{x})$, use MLE with equally weighted training data.

**begin**

   **for** $t = 1, \ldots, T$ **do**

      1. Sample from the current search model, $\tilde{\mathbf{x}}_i^{(t)} \sim p_{\theta^{(t-1)}}(\mathbf{x}), \forall i \in \{1, \ldots, m\}$.

      2. $v_i \leftarrow V(P_{\beta^{(t-1)}}(y \in S \mid \tilde{\mathbf{x}}_i^{(t)})), \forall i \in \{1, \ldots, m\}$.

      3. Fit the updated search model, $p_{\theta^{(t)}}(\mathbf{x})$, using weighted MLE with the samples, $\{\tilde{\mathbf{x}}_i^{(t)}\}_{i=1}^m$, and their corresponding EDA weights, $\{v_i\}_{i=1}^m$.

      4. Compute importance weights for the training data, $w_i \leftarrow p_{\theta^{(t)}}(\mathbf{x}_i)/p_{\theta^{(0)}}(\mathbf{x}_i), i = 1, \ldots, n$.

      5. Retrain the oracle using the re-weighted training data,

$$\beta^{(t)} \leftarrow \arg\max_{\beta \in B} \frac{1}{n} \sum_{i=1}^n w_i \log p_\beta(y_i \mid \mathbf{x}_i).$$

**Output**       :Sequence of search models, $\{p_{\theta^{(t)}}(\mathbf{x})\}_{t=1}^T$, and sequence of samples, $\{(\tilde{\mathbf{x}}_i^{(t)}, \ldots, \tilde{\mathbf{x}}_m^{(t)})\}_{t=1}^T$, from all iterations. One may use these in a number of different ways. For example, one may sample design candidates from the final search model, $p_{\theta^{(T)}}(\mathbf{x})$, or use the most promising candidates among $\{(\tilde{\mathbf{x}}_i^{(t)}, \ldots, \tilde{\mathbf{x}}_m^{(t)})\}_{t=1}^T$, as judged by the appropriate oracle (*i.e.*, corresponding to the iteration at which a candidate was generated).

---

**Algorithm 2: Autofocused Conditioning by Adaptive Sampling (CbAS)**

**Input** : Training data, $\{(\mathbf{x}_i, y_i)\}_{i=1}^n$; oracle model class, $p_\beta(y \mid \mathbf{x})$ with parameters, $\beta$, that can be estimated with MLE; search model class, $p_\theta(\mathbf{x})$, with parameters, $\theta$, that can be estimated with weighted MLE or approximations thereof; maximum number of iterations, $T$; number of samples to generate, $m$; [percentile threshold, $Q = 90$].

**Initialization** : Obtain $p_0(\mathbf{x})$ by fitting to $\{\mathbf{x}_i\}_{i=1}^n$ with the search model class. For the search model, set $p_{\theta^{(0)}}(\mathbf{x}) \leftarrow p_0(\mathbf{x})$. For the oracle, $p_{\beta^{(0)}}(y \mid \mathbf{x})$, use MLE with equally weighted training data. Set $\gamma_0 = -\infty$.

**begin**

  **for** $t = 1, \ldots, T$ **do**

    1. Sample from the current search model, $\tilde{\mathbf{x}}_i^{(t)} \sim p_{\theta^{(t-1)}}(\mathbf{x}), \forall i \in \{1, \ldots, m\}$.

    2. $q_t \leftarrow Q$-th percentile of the oracle expectations of the samples, $\{\mu_\beta(\tilde{\mathbf{x}}_i^{(t)})\}_{i=1}^m$

    3. $\gamma_t \leftarrow \max\{\gamma_{t-1}, q_t\}$

    4. $v_i \leftarrow (p_0(\tilde{\mathbf{x}}_i^{(t)})/p_{\theta^{(t-1)}}(\tilde{\mathbf{x}}_i^{(t)}))P_{\beta^{(t-1)}}(y \geq \gamma_t \mid \tilde{\mathbf{x}}_i^{(t)}), \forall i \in \{1, \ldots, m\}$

    5. Fit the updated search model, $p_{\theta^{(t)}}(\mathbf{x})$, using weighted MLE with the samples, $\{\tilde{\mathbf{x}}_i^{(t)}\}_{i=1}^m$, and their corresponding EDA weights, $\{v_i\}_{i=1}^m$.

    6. Compute importance weights for the training data, $w_i \leftarrow p_{\theta^{(t)}}(\mathbf{x}_i)/p_{\theta^{(0)}}(\mathbf{x}_i), i = 1, \ldots, n$.

    7. Retrain the oracle using the re-weighted training data,

$$\beta^{(t)} \leftarrow \arg\max_{\beta \in B} \frac{1}{n} \sum_{i=1}^n w_i \log p_\beta(y_i \mid \mathbf{x}_i).$$

**Output** : Sequence of search models, $\{p_{\theta^{(t)}}(\mathbf{x})\}_{t=1}^T$, and sequence of samples, $\{(\tilde{\mathbf{x}}_i^{(t)}, \ldots, \tilde{\mathbf{x}}_m^{(t)})\}_{t=1}^T$, from all iterations. One may use these in a number of different ways (see Algorithm 1).

---

**Algorithm 3: Procedure for evaluating MBO algorithms in superconductivity experiments**.
For each MBO algorithm in Tables 1, S2, S3, and S4, the reported scores were the outputs of this procedure, averaged over 10 trials. Recall that $\mu_{\beta^{(t)}}(\mathbf{x}) := \mathbb{E}_{\beta^{(t)}}[y \mid \mathbf{x}]$ denotes the expectation of the oracle model at iteration $t$, while $\mathbb{E}[y \mid \mathbf{x}]$ denotes the ground-truth expectation.

**Input** : Sequence of samples, $\{(\tilde{\mathbf{x}}_i^{(t)}, \ldots, \tilde{\mathbf{x}}_m^{(t)})\}_{t=1}^T$, from each iteration of an MBO algorithm; their oracle expectations, $\{(\mu_{\beta^{(t)}}(\tilde{\mathbf{x}}_i^{(t)}), \ldots, \mu_{\beta^{(t)}}(\tilde{\mathbf{x}}_m^{(t)}))\}_{t=1}^T$; [percentile threshold, $Q = 80$].

**begin**

  **for** $t = 1, \ldots, T$ **do**

    Compute and store $q_t \leftarrow Q$-th percentile of the oracle expectations, $\{\mu_{\beta^{(t)}}(\tilde{\mathbf{x}}_i^{(t)})\}_{i=1}^m$.

  $t_{\text{best}} \leftarrow \arg\max_t q_t$ (pick the best iteration)

  $\mathcal{I} \leftarrow \{i \in \{1, \ldots, m\} : \mu_{\beta^{(t_{\text{best}})}}(\tilde{\mathbf{x}}_i^{(t_{\text{best}})}) \geq q_{t_{\text{best}}}\}$ (pick best samples from best iteration)

  $\mu_{\text{GT,best}} \leftarrow \{\mathbb{E}[y \mid \tilde{\mathbf{x}}_i^{t_{\text{best}}}] : i \in \mathcal{I}\}$

  $\mu_{\text{GT}} \leftarrow (\mathbb{E}[y \mid \tilde{\mathbf{x}}_1^{t_{\text{best}}}], \ldots, \mathbb{E}[y \mid \tilde{\mathbf{x}}_m^{t_{\text{best}}}])$

  $\mu_{\text{oracle}} \leftarrow (\mu_{\beta^{(t_{\text{best}})}}(\tilde{\mathbf{x}}_1^{t_{\text{best}}}), \ldots, \mu_{\beta^{(t_{\text{best}})}}(\tilde{\mathbf{x}}_m^{t_{\text{best}}}))$

  $PCI \leftarrow 100 \cdot \frac{1}{|\mathcal{I}|} \sum_{i \in \mathcal{I}} \mathbf{1}[\mathbb{E}[y \mid \tilde{\mathbf{x}}_i^{t_{\text{best}}}] > \text{maximum label in training data}]$

  $\rho \leftarrow \text{SPEARMAN}(\mu_{\text{GT}}, \mu_{\text{oracle}})$

  $RMSE \leftarrow \text{RMSE}(\mu_{\text{GT}}, \mu_{\text{oracle}})$

**Output** : $\text{median}(\mu_{\text{GT,best}}), \max(\mu_{\text{GT,best}}), PCI, \rho, RMSE$

## S2 Proofs, derivations, and supplementary results

*Proof of Proposition 1.* For any distribution $p_\theta(\mathbf{x})$, if

$$\mathbb{E}_{p_\theta(\mathbf{x})}\left[D_{\mathrm{KL}}(p(y \mid \mathbf{x}) \parallel p_\phi(y \mid \mathbf{x}))\right] \leq \epsilon, \tag{7}$$

then it holds that

$$\mathbb{E}_{p_\theta(\mathbf{x})}\left[|P(y \in S \mid \mathbf{x}) - P_\phi(y \in S \mid \mathbf{x})|^2\right] \leq \mathbb{E}_{p_\theta(\mathbf{x})}\left[\delta(p(y \mid \mathbf{x}), p_\phi(y \mid \mathbf{x}))^2\right] \tag{8}$$

$$\leq \frac{1}{2}\mathbb{E}_{p_\theta(\mathbf{x})}\left[D_{\mathrm{KL}}(p(y \mid \mathbf{x}) \parallel p_\phi(y \mid \mathbf{x}))\right] \tag{9}$$

$$\leq \frac{\epsilon}{2}. \tag{10}$$

where $\delta(p, q)$ is the total variation distance between probability distributions $p$ and $q$, and the second inequality is due to Pinsker's inequality. Finally, applying Jensen's inequality yields

$$\mathbb{E}_{p_\theta(\mathbf{x})}\left[|P(y \in S \mid \mathbf{x}) - P_\phi(y \in S \mid \mathbf{x})|\right] \leq \sqrt{\frac{\epsilon}{2}}. \tag{11}$$

$\square$

### S2.1 Derivation of the descent step to minimize the oracle gap

Here, we derive the descent step of the alternating ascent-descent algorithm described in §3.2. At iteration $t$, given the search model parameters, $\theta^{(t)}$, our goal is to update the oracle parameters according to

$$\beta^{(t)} = \arg\min_{\beta \in B} \mathbb{E}_{p_{\theta^{(t)}}(\mathbf{x})}[D_{\mathrm{KL}}(p(y \mid \mathbf{x}) \parallel p_\beta(y \mid \mathbf{x}))]. \tag{12}$$

Note that

$$\beta^{(t)} = \arg\min_{\beta \in B} \mathbb{E}_{p_{\theta^{(t)}}(\mathbf{x})}\left[\int_{\mathbb{R}} p(y \mid \mathbf{x})\log p(y \mid \mathbf{x})dy - \int_{\mathbb{R}} p(y \mid \mathbf{x})\log p_\beta(y \mid \mathbf{x})dy\right] \tag{13}$$

$$= \arg\max_{\beta \in B} \mathbb{E}_{p_{\theta^{(t)}}(\mathbf{x})}\left[\int_{\mathbb{R}} p(y \mid \mathbf{x})\log p_\beta(y \mid \mathbf{x})dy\right] \tag{14}$$

$$= \arg\max_{\beta \in B} \mathbb{E}_{p_{\theta^{(t)}}(\mathbf{x})}\mathbb{E}_{p(y|\mathbf{x})}[\log p_\beta(y \mid \mathbf{x})]. \tag{15}$$

We cannot query the ground truth, $p(y \mid \mathbf{x})$, but we do have labeled training data, $\{(\mathbf{x}_i, y_i)\}_{i=1}^n$, where $\mathbf{x}_i \sim p_0(\mathbf{x})$ and $y_i \sim p(y \mid \mathbf{x} = \mathbf{x}_i)$ by definition. We therefore leverage importance sampling, using $p_0(\mathbf{x})$ as the proposal distribution, to obtain

$$\beta^{(t)} = \arg\max_{\beta \in B} \mathbb{E}_{p_0(\mathbf{x})}\mathbb{E}_{p(y|\mathbf{x})}\left[\frac{p_{\theta^{(t)}}(\mathbf{x})}{p_0(\mathbf{x})}\log p_\beta(y \mid \mathbf{x})\right]. \tag{16}$$

Finally, we instantiate an importance sampling estimate of the objective in Equation 16 with our labeled training data, to get a practical oracle parameter update,

$$\beta^{(t)} = \arg\max_{\beta \in B} \frac{1}{n}\sum_{i=1}^n \frac{p_{\theta^{(t)}}(\mathbf{x}_i)}{p_0(\mathbf{x}_i)}\log p_\beta(y_i \mid \mathbf{x}_i). \tag{17}$$

This update is equivalent to fitting the oracle parameters, $\beta^{(t)}$, by performing weighted MLE with the labeled training data, $\{(\mathbf{x}_i, y_i)\}_{i=1}^n$, and corresponding weights, $\{w_i\}_{i=1}^n$, where $w_i := p_{\theta^{(t)}}(\mathbf{x}_i)/p_0(\mathbf{x}_i)$.

### S2.2 Variance of importance weights

The importance-sampled estimate of the log-likelihood used to retrain the oracle (Equation 17) is unbiased, but may have high variance due to the variance of the importance weights. To assess the reliability of the importance-sampled estimate, alongside the effective sample size described in §3.3, one can also monitor confidence intervals on some loss of interest. Let $\mathcal{L}_\beta : \mathcal{X} \times \mathbb{R} \to \mathbb{R}$ denote a pertinent loss function induced by the oracle parameters, $\beta$, (*e.g.*, the squared error $\mathcal{L}_\beta(\mathbf{x}, y) = (\mathbb{E}_\beta[y \mid \mathbf{x}] - y)^2$). The following observation is due to Chebyshev's inequality.

**Proposition S2.1.** *Suppose that $\mathcal{L}_\beta : \mathcal{X} \times \mathbb{R} \to \mathbb{R}$ is a bounded loss function, such that $|\mathcal{L}_\beta(\mathbf{x}, y)| \leq L$ for all $\mathbf{x}, y$, and that $p_\theta \ll p_0$. Let $\{(\mathbf{x}_i, y_i)\}_{i=1}^n$ be labeled training data such that the $\mathbf{x}_i \sim p_0(\mathbf{x})$ are drawn independently and $y_i \sim p(y \mid \mathbf{x} = \mathbf{x}_i)$ for each $i$. For any $\delta \in (0, 1]$ and any $n > 0$, with probability at least $1 - \delta$ it holds that*

$$\left| \mathbb{E}_{p_\theta(\mathbf{x})} \mathbb{E}_{p(y|\mathbf{x})}[\mathcal{L}_\beta(\mathbf{x}, y)] - \frac{1}{n} \sum_{i=1}^n \frac{p_\theta(\mathbf{x}_i)}{p_0(\mathbf{x}_i)} \mathcal{L}_\beta(\mathbf{x}_i, y_i) \right| \leq L \sqrt{\frac{d_2(p_\theta \,\|\, p_0)}{n\delta}} \tag{18}$$

*where $d_2$ is the exponentiated Rényi-2 divergence, i.e., $d_2(p_\theta \,\|\, p_0) = \mathbb{E}_{p_0(\mathbf{x})}[(p_\theta(\mathbf{x})/p_0(\mathbf{x}))^2]$.*

*Proof.* We use the following lemma to bound the variance of the importance sampling estimate of the loss. Chebyshev's inequality then yields the desired result. □

**Lemma S2.1** (Adaptation of Lemma 4.1 in Metelli et al. (2018) [50]). *Under the same assumptions as Proposition S2.1, the joint distribution $p_\theta(\mathbf{x})p(y \mid \mathbf{x})$ is absolutely continuous with respect to the joint distribution $p_0(\mathbf{x})p(y \mid \mathbf{x})$. Then for any $n > 0$, it holds that*

$$\mathrm{Var}_{p_\theta(\mathbf{x})p(y|\mathbf{x})} \left[ \frac{1}{n} \sum_{i=1}^n \frac{p_\theta(\mathbf{x}_i)}{p_0(\mathbf{x}_i)} \mathcal{L}_\beta(\mathbf{x}_i, y_i) \right] \leq \frac{1}{n} L^2 d_2(p_\theta \| p_0). \tag{19}$$

One can use Proposition S2.1 to construct a confidence interval on, for example, the expected squared error between the oracle and the ground-truth values with respect to $p_\theta(\mathbf{x})$, using the labeled training data on hand. The Rényi divergence can be estimated using, for example, the plug-in estimate $(1/n) \sum_{i=1}^n (p_\theta(\mathbf{x}_i)/p_0(\mathbf{x}_i))^2$. While the bound, $L$, on $\mathcal{L}_\beta$ may be restrictive in general, for any given application one may be able to use domain-specific knowledge to estimate $L$. For example, in designing superconducting materials with maximized critical temperature, one can use an oracle architecture whose outputs are non-negative and at most some plausible maximum value $M$ (in degrees Kelvin) according to superconductivity theory; one could then take $L = M^2$ for squared error loss. Computing a confidence interval at each iteration of a design procedure then allows one to monitor the error of the retrained oracle.

Monitoring such confidence intervals, or the effective sample size, is most likely to be useful for design procedures that do not have in-built mechanisms for restricting the movement of the search distribution away from the training distribution. Various algorithmic interventions are possible—one could simply terminate the procedure if the error bounds, or effective sample size, surpass some threshold, or one could decide not to retrain the oracle for that iteration. For simplicity and clarity of exposition, we did not use any such interventions in this paper, but we mention them as potential avenues for further improving autofocusing in practice. Note that 1) the bound in Proposition S2.1 is only useful if the importance weight variance is finite, and 2) estimating the bound itself requires use of the importance weights, and thus may also be susceptible to high variance. It may therefore be advantageous to use a liberal criterion for any interventions.

**CbAS naturally controls the importance weight variance.** Design procedures that leverage a trust region can naturally bound the variance of the importance weights. For instance, CbAS [11], developed in the context of an oracle with fixed parameters, $\beta$, proposes estimating the training distribution conditioned on $S$ as the search model:

$$p_\theta(\mathbf{x}) = p_0(\mathbf{x} \mid S) = P_\beta(S \mid \mathbf{x})p_0(\mathbf{x})/P_0(S), \tag{20}$$

where $P_0(S) = \int P_\beta(S \mid \mathbf{x})p_0(\mathbf{x})dx$. This prescribed search model yields the following importance weight variance.

**Proposition S2.2.** *For $p_\theta(\mathbf{x}) = p_0(\mathbf{x} \mid S)$, it holds that*

$$\mathrm{Var}_{p_0(\mathbf{x})} \left( \frac{p_\theta(\mathbf{x})}{p_0(\mathbf{x})} \right) = \frac{1}{P_0(S)} - 1. \tag{21}$$

That is, so long as $S$ has non-neglible mass under data drawn from the training distribution, $p_0(\mathbf{x})$, we have reasonable control on the variance of the importance weights. Of course, if $P_0(S)$ is too small, this bound is not useful, but to have any hope of successful data-driven design it is only reasonable to expect this quantity to be non-negligible. This is similar to the experimental requirement, in directed evolution for protein design, that the initial data exhibit some "minimal functionality" with regards to the property of interest [3].

*Proof.* The variance of the importance weights can be written as

$$\text{Var}_{p_0(\mathbf{x})}\left(\frac{p_0(\mathbf{x} \mid S)}{p_0(\mathbf{x})}\right) = d_2(p_0(\mathbf{x} \mid S)||p_0(\mathbf{x})) - 1, \tag{22}$$

where $d_2(p_0(\mathbf{x} \mid S)||p_0(\mathbf{x})) = \mathbb{E}_{p_0(\mathbf{x})}[(p_0(\mathbf{x} \mid S)/p_0(\mathbf{x}))^2]$ is the exponentiated Rényi-2 divergence. Then we have

$$\text{Var}_{p_0(\mathbf{x})}\left(\frac{p_\theta(\mathbf{x})}{p_0(\mathbf{x})}\right) = d_2(p_0(\mathbf{x} \mid S)||p_0(\mathbf{x})) - 1 = \frac{1}{p_0(S)} - 1, \tag{23}$$

where the second equality is due to the property in Example 1 of [51]. □

This variance yields the following expression for the population version of the effective sample size:

$$n_e^* := \frac{n\mathbb{E}_{p_0(x)}\left[p_\theta(\mathbf{x})/p_0(\mathbf{x})\right]^2}{\mathbb{E}_{p_0(x)}\left[(p_\theta(\mathbf{x})/p_0(\mathbf{x}))^2\right]} = \frac{n}{\mathbb{E}_{p_0(x)}\left[(p_\theta(\mathbf{x})/p_0(\mathbf{x}))^2\right]} = nP_0(S). \tag{24}$$

Furthermore, CbAS proposes an iterative procedure to estimate $p_\theta(\mathbf{x})$. At iteration $t$, the algorithm seeks a variational approximation to $p^{(t)}(\mathbf{x}) \propto P_\beta(S^{(t)} \mid \mathbf{x})p_0(\mathbf{x})$, where $S^{(t)} \supseteq S$. Since $P_0(S^{(t)} \mid \mathbf{x}) \geq P_0(S \mid \mathbf{x})$, the expressions above for the importance weight variance and effective sample size for the final search model prescribed by CbAS translate into upper and lower bounds, respectively, on the importance weight variance and effective sample size for the distributions $p^{(t)}(\mathbf{x})$ prescribed at each iteration.

## S3   An illustrative example

### S3.1   Experimental details

**Ground truth and oracle.**   For the ground-truth function $f : \mathbb{R} \rightarrow \mathbb{R}^+$, we used the sum of the densities of two Gaussian distributions, $\mathcal{N}_1(5, 1)$ and $\mathcal{N}_2(7, 0.25)$. For the expectation of the oracle model, $\mu_\beta(\mathbf{x}) := \mathbb{E}_\beta[y \mid \mathbf{x}]$, we used kernel ridge regression with a radial basis function kernel as implemented in `scikit-learn`, with the default values for all hyperparameters. The variance of the oracle model, $\sigma_\beta^2 := \text{Var}_\beta[y \mid \mathbf{x}]$, was set to the mean squared error between $\mu_\beta(\mathbf{x})$ and the training data labels, as estimated with 4-fold importance-weighted cross-validation when autofocusing [24].

**MBO algorithm.**   We used CbAS as follows. At iteration $t = 1, \ldots, 100$, similar to [11], we used the relaxed constraint set $S^{(t)} = \{y : y \geq \gamma_t\}$ where $\gamma_t$ was the $t^{th}$ percentile of the oracle expectation, $\mu_\beta(\mathbf{x})$, when evaluated over $\mathbf{x} \in [0, 10]$. At the final iteration, $t = 100$, the constraint set is equivalent to the design goal of maximizing the oracle expectation, $S^{(100)} = S = \{y : y \geq \max_\mathbf{x} \mu_\beta(\mathbf{x})\}$, which is the oracle-based proxy to maximizing the ground-truth function, $f(\mathbf{x})$. At each iteration, we used numerical quadrature (`scipy.integrate.quad`) to compute the search model,

$$p^{(t)}(\mathbf{x}) = \frac{P_{\beta^{(t)}}(y \in S^{(t)} \mid \mathbf{x})\, p_0(\mathbf{x})}{\int_\mathcal{X} P_{\beta^{(t)}}(y \in S^{(t)} \mid \mathbf{x})\, p_0(\mathbf{x})}. \tag{25}$$

Numerical integration enabled us to use CbAS without a parametric search model, which otherwise would have been used to find a variational approximation to this distribution [11]. We also used numerical integration to compute the value of the model-based design objective (Equation 1) achieved by the final search model, both with and without autofocusing.

### S3.2   Additional plots and discussion

For all tested settings of the variance of the training distribution, $\sigma_0^2$, and the variance of the label noise, $\sigma_\epsilon^2$, autofocusing yielded positive improvement to the model-based design objective (Equation 1) on average over 50 trials (Figure 2). For a more comprehensive understanding of the effects of autofocusing, here we pinpoint specific trials where autofocusing decreased the objective, compared to a naive approach with a fixed oracle. Such trials were rare, and occurred in regimes where one

would not reasonably expect autofocusing to provide a benefit. In particular, as discussed in §5.1, such regimes include when $\sigma_0^2$ is too small, such that training data are unlikely to be close to the global maximum, and when $\sigma_0^2$ is too large, such that the training data already include points around the global maximum and a fixed oracle should be suitable for successful design. Similarly, when the label noise variance, $\sigma_\epsilon^2$, is too large, the training data are no longer informative and no procedure should hope to perform well systematically. We now walk through each of these regimes.

When $\sigma_0^2$ was small and there was no label noise, we observed a few trials where the final search model placed less mass under the global maximum with autofocusing than without. This effect was due to increased standard deviation of the autofocused oracle, induced by high variance of the importance weights (Figure S1a). When $\sigma_0^2$ was small and $\sigma_\epsilon^2$ was extremely large, a few trials yielded lower final objectives with autofocusing by insignificant margins; in such cases, the label noise was overwhelming enough that the search model did not move much anyway, either with or without autofocusing (Figure S1b). Similarly, when $\sigma_0^2$ was large and there was no label noise, a few trials yielded lower final objectives with autofocusing than without, by insignificant margins (Figure S1c).

Interestingly, when the variances of both the training distribution and label noise were high, autofocusing yielded positive improvement for all trials. In this regime, by encouraging the oracle to fit most accurately to the points with the highest labels, autofocusing resulted in search models with greater mass under the global maximum than the fixed-oracle approach, which was more influenced by the extreme label noise (Figure S1d).

As discussed in §5.1, in practice it is often the case that 1) practitioners can collect reasonably informative training data for the application of interest, such that some exceptional examples are measured (corresponding to sufficiently large $\sigma_0^2$), and 2) there is always label noise, due to measurement error (corresponding to non-zero $\sigma_\epsilon^2$). Thus, we expect many design applications in practice to fall in the intermediate regime where autofocusing tends to yield positive improvements over a fixed-oracle approach (Figure 2, Table 1).

## S4   Designing superconductors with maximal critical temperature

### S4.1   Experimental details

**Pre-processing.**   Each of the $21,263$ materials in the superconductivity data from [44] is represented by a vector of eighty-one real-valued features. We zero-centered and normalized each feature to have unit variance.

**Ground-truth model.**   To construct the model of the ground-truth expectation, $\mathbb{E}[y \mid \mathbf{x}]$, we fit gradient-boosted regression trees using `xgboost` and the same hyperparameters reported in [44], which selected them using grid search. The one exception was that we used 200 trees instead of 750 trees, which yielded a hold-out root mean squared error (RMSE) of $9.51$ compared to the hold-out RMSE of $9.5$ reported in [44]. To remove collinear features noted in [44], we also performed feature selection by thresholding `xgboost`'s in-built feature weights, which quantifies how many times a feature is used to split the data across all trees. We kept the sixty most important features according to this score, which decreased the hold-out RMSE from $9.51$ when using all the features to $9.45$. The resulting input space for design was then $\mathcal{X} = \mathbb{R}^{60}$.

**Training distribution.**   To construct the training distribution, we selected the $17,015$ points from the dataset whose ground-truth expectations were below the $80^{th}$ percentile (equivalent to $73.8$ degrees Kelvin, compared to the maximum of $138.3$ degrees Kelvin in the full dataset). We used MLE with these points to fit a full-rank multivariate normal, which served as the training distribution, $p_0(\mathbf{x})$, from which we drew $n = 17,015$ simulated training points, $\{\mathbf{x}_i\}_{i=1}^n$, for each trial. For each $\mathbf{x}_i$ we drew one sample, $y_i \sim \mathcal{N}(\mathbb{E}[y \mid \mathbf{x}_i], 1)$, to obtain a noisy ground-truth label. This training distribution produced simulated training points with a distribution of ground-truth expectations, $\mathbb{E}[y \mid \mathbf{x}]$, reasonably comparable to that of the points from the original dataset (Figure S2, left panel).

**Oracle.**   For the oracle, we trained an ensemble of three neural networks to maximize log-likelihood according to the method described in [46] (without adversarial examples). Each model in the ensemble had the architecture `Input(60)` $\rightarrow$ `Dense(100)` $\rightarrow$ `Dense(100)` $\rightarrow$ `Dense(100)` $\rightarrow$

(a) Example trial with low-variance training distribution and no label noise, $(\sigma_0, \sigma_\epsilon) = (1.6, 0)$.

(b) Example trial with low-variance training distribution and high label noise, $(\sigma_0, \sigma_\epsilon) = (1.6, 0.38)$.

(c) Example trial with high-variance training distribution and no label noise $(\sigma_0, \sigma_\epsilon) = (2.2, 0)$.

(d) Example trial with high-variance training distribution and high label noise $(\sigma_0, \sigma_\epsilon) = (2.2, 0.38)$.

Figure S1: Examples of regimes where autofocus (AF) sometimes yielded lower final objectives than without (non-AF). Each row shows snapshots of CbAS in a different experimental regime, from initialization (leftmost panel), to an intermediate iteration (middle panel), to the final iteration (rightmost panel). As in Figure 1, the vertical axis represents both $y$ values (for the oracle and ground truth) and probability density values (of the training distribution, $p_0(\mathbf{x})$, and search distributions, $p_{\theta^{(t)}}(\mathbf{x})$). Shaded envelopes correspond to $\pm 1$ standard deviation of the oracles, $\sigma_{\beta^{(t)}}$, with the oracle expectations, $\mu_{\beta^{(t)}}(\mathbf{x})$, shown as a solid line. Greyscale of training points corresponds to their importance weights used in autofocusing. In the rightmost panels, for easy visualization of the final search models achieved with and without AF, the stars and dotted horizontal lines indicate the ground-truth values corresponding to the points of maximum density.

Figure S2: Training distribution and initial oracle for designing superconductors. Simulated training data were generated from a training distribution, $p_0(\mathbf{x})$, which was a multivariate Gaussian fit to data points with ground-truth expectations below the $80^{th}$ percentile. The left panel shows histograms of the ground-truth expectations of these original data points, and the ground-truth expectations of simulated training data. The right panel illustrates the error of an initial oracle used in the experiments, by plotting the ground-truth and predicted labels of $10,000$ test points drawn from the training distribution. The RMSE here was $7.31$.

`Dense(100)` $\rightarrow$ `Dense(10)` $\rightarrow$ `Dense(2)`, with `elu` nonlinearities everywhere except for linear output units. Out of the range of hidden layer numbers and sizes we tested, this architecture minimized RMSE on held-out data. Each model was trained using Adam [52] with a learning rate of $5 \times 10^{-4}$ for a maximum of $2000$ epochs, with a batch size of $64$ and early stopping based on the log-likelihood of a validation set. Across the $10$ trials, the initial oracles had hold-out RMSEs between $6.95$ and $7.40$ degrees Kelvin (Figure S2, right panel).

**Autofocusing.** During autofocusing, each model in the oracle ensemble was retrained with the re-weighted training data, using the same optimization hyperparameters as the initial oracle, except early stopping was based on the re-weighted log-likelihood of the validation set. For the results in Table 1, to help control the variance of the importance weights, we flattened the importance weights to $w_i^\alpha$ where $\alpha = 0.2$ [24] and also self-normalized them [26]. We found that autofocusing yielded similarly widespread benefits for a wide range of values of $\alpha$, including $\alpha = 1$, which corresponds to a "pure" autofocusing strategy without variance control (Table S2).

**MBO algorithms.** Here, we provide a brief description of the different MBO algorithms used in the superconductivity experiments (Tables 1, S2, S3, S4, Figures S3 and S4). Wherever applicable, in parentheses we anchor these descriptions in the notation and procedure of Algorithm 1.

- *Design by Adaptive Sampling (DbAS)* [11]. A basic EDA that anneals a sequence of relaxed constraint sets, $S^{(t)}$, such $S^{(t)} \supseteq S^{(t+1)} \supseteq S$, to iteratively solve the oracle-based MBD problem (Equation 2). (At iteration $t$, DbAS uses $V(\tilde{\mathbf{x}}_i^{(t)}) = P_{\beta^{(t-1)}}(y \in S^{(t)} \mid \tilde{\mathbf{x}}_i^{(t)})$.)

- *Conditioning by Adaptive Sampling (CbAS)* [11]. Seeks to estimate the training distribution conditioned on the desired constraint set $S$ (Equation 20). Similar mechanistically to DbAS, as it involves constructing a sequence of relaxed constraint sets, but also incorporates an implicit trust region based on the training distribution. (At iteration $t$, CbAS uses $V(\tilde{\mathbf{x}}_i^{(t)}) = (p_0(\tilde{\mathbf{x}}_i^{(t)})/p_{\theta^{(t-1)}}(\tilde{\mathbf{x}}_i^{(t)}))P_{\beta^{(t-1)}}(y \in S^{(t)} \mid \tilde{\mathbf{x}}_i^{(t)})$. See Algorithm 2; non-autofocused CbAS excludes Steps 6 and 7.)

- *Reward-Weighted Regression (RWR)* [23]. An EDA used in the reinforcement learning community. (At iteration $t$, RWR uses $V(\tilde{\mathbf{x}}_i^{(t)}) = v_i'/\sum_{j=1}^m v_j'$, where $v_i' = \exp(\gamma \mathbb{E}_{\beta^{(t-1)}}[y \mid \tilde{\mathbf{x}}_i^{(t)}])$) and $\gamma > 0$ is a hyperparameter).

- *"Feedback" Mechanism (FB)* [17]. A heuristic version of CbAS, which maintains samples from previous iterations to prevent the search model from changing too rapidly. (At Step 3 in Algorithm 1, FB uses samples from the current iteration with oracle expectations that surpass some percentile threshold, along with a subset of promising samples from previous iterations.)

Table S2: Designing superconducting materials. Same experiments and caption as Table 1, except with $\alpha = 1$ (no flattening of the importance weights to control variance). We ran six different MBO methods, each with and without autofocusing. For each method, we extracted those samples with oracle expectations above the $80^{th}$ percentile and computed their ground-truth expectations. We report the median and maximum of those ground-truth expectations (both in degrees K), their percent chance of improvement (PCI, in percent) over the maximum label in the training data, as well as the Spearman correlation ($\rho$) and root mean squared error (RMSE, in degrees K) between the oracle and ground-truth expectations. Each reported score is averaged over 10 trials, where, in each trial, a different training set was sampled from the training distribution. "Mean Diff." is the average difference between the score when using autofocusing compared to not. Bold values with one star (*) and two stars (**), respectively, mean $p$-value $< 0.05$ and $< 0.01$ from a two-sided Wilcoxon signed-rank test on the 10 paired score differences. For all scores but RMSE, a higher value means autofocusing yielded better results (as indicated by the arrow $\uparrow$); for RMSE, the opposite is true (as indicated by the arrow $\downarrow$).

| | Median ↑ | Max ↑ | PCI ↑ | $\rho$ ↑ | RMSE ↓ | Median ↑ | Max ↑ | PCI ↑ | $\rho$ ↑ | RMSE ↓ |
|---|---|---|---|---|---|---|---|---|---|---|
| | | | **CbAS** | | | | | **DbAS** | | |
| Original | 51.5 | 103.8 | 0.11 | 0.05 | 17.2 | 57.0 | 98.4 | 0.11 | 0.01 | 29.6 |
| Autofocused | 73.2 | 116.0 | 2.29 | 0.56 | 12.8 | 69.4 | 109.9 | 0.68 | 0.01 | 27.4 |
| Mean Diff. | **21.8**** | **12.2**** | **2.18**** | **0.51**** | **-4.4**** | **12.4**** | **11.5**** | **0.58**** | 0.01 | -2.2 |
| | | | **RWR** | | | | | **FB** | | |
| Original | 43.4 | 102.0 | 0.05 | 0.92 | 7.4 | 9.2 | 100.6 | 0.14 | 40.09 | 17.5 |
| Autofocused | 68.5 | 113.4 | 1.34 | 0.63 | 14.2 | 63.4 | 110.8 | 0.63 | 0.49 | 11.2 |
| Mean Diff. | **25.1**** | **11.5**** | **1.30**** | **-0.29**** | **6.8**** | **14.2**** | **10.2*** | **0.50**** | **0.40**** | **-6.3**** |
| | | | **CEM-PI** | | | | | **CMA-ES** | | |
| Original | 34.5 | 55.8 | 0.00 | -0.16 | 148.3 | 42.1 | 69.4 | 0.00 | 0.27 | 27493.2 |
| Autofocused | 59.5 | 89.1 | 0.39 | 0.02 | 46.9 | 45.1 | 70.7 | 0.00 | 0.50 | 27944.9 |
| Mean Diff. | **25.0*** | **33.3*** | **0.39*** | 0.18 | **-101.4**** | 3.1 | 1.2 | 0 | **0.22*** | 451.7 |

- *Cross-Entropy Method with Probability of Improvement (CEM-PI)* [11]. A baseline EDA that uses the cross-entropy method [21, 22] to maximize the probability of improvement, an acquisition function commonly used in Bayesian optimization [32]. (At iteration $t$, CEM-PI uses $V(\tilde{\mathbf{x}}_i^{(t)}) = \mathbf{1}[P_{\beta^{(t)}}(y \geq y_{\max} \mid \tilde{\mathbf{x}}_i^{(t)}) \geq \gamma_t]$, where $y_{\max}$ is the maximum label observed in the training data, and, following the cross-entropy method, $\gamma_t$ is some percentile of the probabilities of improvement according to the oracle, $\{P_{\beta^{(t)}}(y \geq y_{\max} \mid \tilde{\mathbf{x}}_i^{(t)})\}_{i=1}^m$.)
- *Covariance Matrix Adaptation Evolution Strategy (CMA-ES)* [47]. A state-of-the-art EDA developed for the special case of multivariate Gaussian search models. We used it to maximize the probability of improvement according to the oracle, $P_{\beta^{(t)}}(y \geq y_{\max} \mid \tilde{\mathbf{x}}_i^{(t)})$.

CbAS, DbAS, FB, and CEM-PI all have hyperparameters corresponding to a percentile threshold (for CbAS and DbAS, this is used to construct the relaxed constraint sets). We set this hyperparameter to 90 for all these methods. For RWR, we set $\gamma = 0.01$, and for CMA-ES, we set the step size hyperparameter to $\sigma = 0.01$.

## S4.2 Additional experiments

**Importance weight variance control.** To see how much importance weight variance affects autofocusing, we conducted the same experiments as Table 1, except without flattening the weights to reduce variance (Table S2). For CbAS, DbAS, RWR, FB, and CEM-PI, autofocusing without variance control yielded statistically significant improvements to the majority of scores, though with somewhat lesser effect sizes than in Table 1 when the weights were flattened with $\alpha = 0.2$. For CMA-ES, the only significant improvement autofocusing rendered was to the Spearman correlation between the oracle and the ground-truth expectations. Note that CMA-ES is a state-of-the-art method for optimizing a given objective with a multivariate Gaussian search model [47], which likely led to liberal movement of the search model away from the training distribution and therefore high importance weight variance.

**Oracle capacity.** To see how different oracle capacities affect the improvements gained from autofocusing, we ran the same experiments as Table 1 with two different oracle architectures. One architec-

(a) CbAS.

(b) DbAS.

(c) RWR.

(d) FB.

Figure S3: Designing superconducting materials. Trajectories of different MBO algorithms run without (left) and with autofocusing (right), on one example trial used to compute Table 1. At each iteration, we extract the samples with oracle expectations greater than the $80^{th}$ percentile. For these samples, solid lines give the median oracle (green) and ground-truth (indigo) expectations. The shaded regions capture 70 and 95 percent of these quantities. The RMSE at each iteration is between the oracle and ground-truth expectations of all samples. The horizontal axis is sorted by increasing $80^{th}$ percentile of oracle expectations (*i.e.*, the samples plotted at iteration 1 are from the iteration whose $80^{th}$ percentile of oracle expectations was lowest). This ordering exposes the trend of whether the oracle expectations of samples were correlated to their ground-truth expectations. Two more algorithms are shown in Figure S4.

(a) CEM-PI.

(b) CMA-ES.

Figure S4: Designing superconducting materials. Continuation of Figure S3.

ture had higher capacity than the original oracle (hidden layer sizes of (200, 200, 100, 100, 10) compared to (100, 100, 100, 100, 10); Table S3), and one one had lower capacity (hidden layer sizes of (100, 100, 10); Table S4).

Table S3: Designing superconducting materials. Same experiments and caption as Table 1, except using an oracle architecture with hidden layers $200 \rightarrow 200 \rightarrow 100 \rightarrow 100 \rightarrow 10$. We ran six different MBO methods, each with and without autofocusing. For each method, we extracted those samples with oracle expectations above the $80^{th}$ percentile and computed their ground-truth expectations. We report the median and maximum of those ground-truth expectations (both in degrees K), their percent chance of improvement (PCI, in percent) over the maximum label in the training data, as well as the Spearman correlation ($\rho$) and root mean squared error (RMSE, in degrees K) between the oracle and ground-truth expectations. Each reported score is averaged over 10 trials, where, in each trial, a different training set was sampled from the training distribution. "Mean Diff." is the average difference between the score when using autofocusing compared to not. Bold values with one star (*) and two stars (**), respectively, mean $p$-value $< 0.05$ and $< 0.01$ from a two-sided Wilcoxon signed-rank test on the 10 paired score differences. For all scores but RMSE, a higher value means autofocusing yielded better results (as indicated by the arrow $\uparrow$); for RMSE, the opposite is true (as indicated by the arrow $\downarrow$).

| | Median ↑ | Max ↑ | PCI ↑ | $\rho$ ↑ | RMSE ↓ | Median ↑ | Max ↑ | PCI ↑ | $\rho$ ↑ | RMSE ↓ |
|---|---|---|---|---|---|---|---|---|---|---|
| | | | CbAS | | | | | DbAS | | |
| Original | 48.3 | 100.8 | 0.05 | 0.03 | 19.6 | 55.3 | 98.6 | 0.025 | -0.02 | 32.1 |
| Autofocused | 79.0 | 119.4 | 4.35 | 0.55 | 13.5 | 81.6 | 113.3 | 5.33 | 0.01 | 27.0 |
| Mean Diff. | **30.7**** | **18.6**** | **4.30**** | **0.52**** | **-6.1**** | **26.4**** | **14.8**** | **5.30**** | 0.03 | -5.1 |
| | | | RWR | | | | | FB | | |
| Original | 36.5 | 81.3 | 0.00 | -0.24 | 55.5 | 47.8 | 101.5 | 0.09 | 0.06 | 18.3 |
| Autofocused | 73.4 | 114.8 | 2.05 | 0.72 | 12.7 | 63.5 | 113.1 | 0.58 | 0.58 | 10.7 |
| Mean Diff. | **36.9**** | **33.4**** | **2.05**** | **0.97**** | **-42.8**** | **15.7**** | **11.7**** | **0.49**** | **0.51**** | **-7.5**** |
| | | | CEM-PI | | | | | CMA-ES | | |
| Original | 48.2 | 58.3 | 0.00 | 0.09 | 271.4 | 39.0 | 63.1 | 0.00 | 0.26 | 6774.6 |
| Autofocused | 64.5 | 84.1 | 0.48 | -0.14 | 61.07 | 53.1 | 79.0 | 0.01 | 0.48 | 10183.7 |
| Mean Diff. | 16.3 | **25.9*** | 0.48 | -0.22 | **-210.3** | **14.1*** | **15.9*** | 0.01 | 0.23 | 3409.1 |

Table S4: Designing superconducting materials. Same experiments and caption as Table 1, except using an oracle architecture with hidden layers $100 \rightarrow 100 \rightarrow 10$. We ran six different MBO methods, each with and without autofocusing. For each method, we extracted those samples with oracle expectations above the $80^{th}$ percentile and computed their ground-truth expectations. We report the median and maximum of those ground-truth expectations (both in degrees K), their percent chance of improvement (PCI, in percent) over the maximum label in the training data, as well as the Spearman correlation ($\rho$) and root mean squared error (RMSE, in degrees K) between the oracle and ground-truth expectations. Each reported score is averaged over 10 trials, where, in each trial, a different training set was sampled from the training distribution. "Mean Diff." is the average difference between the score when using autofocusing compared to not. Bold values with one star (*) and two stars (**), respectively, mean $p$-value $< 0.05$ and $< 0.01$ from a two-sided Wilcoxon signed-rank test on the 10 paired score differences. For all scores but RMSE, a higher value means autofocusing yielded better results (as indicated by the arrow $\uparrow$); for RMSE, the opposite is true (as indicated by the arrow $\downarrow$).

| | Median ↑ | Max ↑ | PCI ↑ | $\rho$ ↑ | RMSE ↓ | Median ↑ | Max ↑ | PCI ↑ | $\rho$ ↑ | RMSE ↓ |
|---|---|---|---|---|---|---|---|---|---|---|
| | | | CbAS | | | | | DbAS | | |
| Original | 0.06 | 46.8 | 98.5 | -0.03 | 23.8 | 0.02 | 56.3 | 97.7 | 0.00 | 37.0 |
| Autofocused | 1.4 | 67.0 | 114.3 | 0.52 | 13.0 | 1.3 | 72.5 | 108.4 | 0.04 | 27.6 |
| Mean Diff. | **1.3**** | **20.2**** | **15.8**** | **0.55**** | **-10.9**** | **1.3**** | **16.2**** | **10.7**** | 0.03 | **-9.4**** |
| | | | RWR | | | | | FB | | |
| Original | 0.00 | 30.9 | 76.8 | -0.33 | 83.5 | 0.04 | 47.2 | 100.4 | 0.02 | 19.9 |
| Autofocused | 0.68 | 66.0 | 112.6 | 0.57 | 18.3 | 0.43 | 58.2 | 111.4 | 0.50 | 12.3 |
| Mean Diff. | **0.68**** | **35.1**** | **35.8**** | **0.90**** | **-65.2**** | **0.40**** | **11.0**** | **11.0**** | **0.48**** | **-7.6**** |
| | | | CEM-PI | | | | | CMA-ES | | |
| Original | 0.00 | 36.3 | 46.2 | -0.01 | 382.4 | 0.00 | 36.9 | 62.3 | 0.10 | 9587.1 |
| Autofocused | 0.04 | 53.9 | 71.3 | -0.04 | 210.8 | 0.00 | 43.9 | 80.0 | 0.29 | 40858.6 |
| Mean Diff. | 0.04 | 17.6 | **25.1*** | -0.03 | -171.6 | 0 | **7.0*** | **17.7*** | **0.19**** | **31271.5**** |