[Reviews · NeurIPS 2020]

Review 1

Summary and Contributions: The authors seek to address a central challenge in what they refer to as oracle-based design (design seeking to optimise a property through the use of a regression model), which is that in many situations the optimum one is seeking has never been seen and hence must necessarily lie outside of the training data. Thus the goal of design requires using the model in a regime where it is unreliable. The proposed strategy is to iteratively train the oracle (predictive model) by re- the training data to “autofocus” on the region of design candidates. This is achieved by extending an Estimation of Distribution Algorithm (EDA) to consider two coupled objectives which not only optimise the parameters of the search model but also minimise the “oracle gap”.

Strengths: The paper is seeking to address a fundamental problem in model based design -- an extremely exciting area of research with widespread application in developing new materials, chemical compounds, biological sequences and beyond. As such, this work has the potential to be very high impact and it will be exciting to see it applied to real design problems. Furthermore, the approach taken here is principled, theoretically grounded and versatile, being clear how to implement it in different contexts. To my knowledge, while the approach touches on numerous ideas from disparate fields, it is original.

Weaknesses: For such a thoughtful paper, I found the Related Work section underwhelming. The authors emphasise that there is no work seeking to update the oracle, which is important and argued clearly but discussion of areas of application and also the connection to other approaches is rather minimal. For instance, protein design is mentioned multiple times as a possible area of application but it is unclear how effective this approach will be in overcoming the major challenges currently being faced in this field. It’s a difficult literature to navigate but perhaps this repository may be of some use https://github.com/yangkky/Machine-learning-for-proteins. These are not topics I know particularly well but I would have thought there would also be interesting connections to be made to, for instance, out of distribution generalisation, small area estimation and model misspecification. Also, while the arguments and methodology are stated clearly, the extent of the practical gains from this approach are not so transparent, in the sense that it will lie in the details of the specific design problem and in the interaction with the model.

Correctness: Yes

Clarity: It was a pleasure to read. The arguments are clear, convincing and thought provoking

Relation to Prior Work: The distinction from previous work is clear, although as I mentioned above, I feel like more could have been said here, particularly with regard to what is common to other approaches.

Reproducibility: Yes

Additional Feedback:


Review 2

Summary and Contributions: The problem of data-driven design is an area of intense interest especially in chemistry and biology. A scenario often encountered in this setting is when a (regression) model is trained based on some collected data (expensive), and new samples are desired that somehow optimize the properties of the input. However, a challenge often encountered (though not always acknowledged) is that the trained model is misspecified and starts to become inaccurate as the inputs deviate from those present in the training data. Many search algorithms that query such models define a "trust region" (which can be soft or hard) and only accept designed inputs if they fall within that region. In this paper, the authors tackle this problem with a different approach: they ask if the regression model can be retrained based the data that was already collected, as the "search model" is iteratively updated and moves through the input space, to "auto-focus" (i.e. emphasize the more relevant bits of data in retraining the model) and improve the optimization process. The authors propose a coupled training scheme for the search model and the regression model until convergence, and show that they can bound the expected deviation from the ground-truth (i.e. avoiding pathological maxima).

Strengths: The problem that this study is approaching is very relevant and (with certain advances) can have impact across multiple domains. I find the autofocusing approach appealing, and the empirical evidence that the authors provide suggests that it can be useful in settings that a good probabilistic search model can be specified. Theoretical justifications are provided throughout. It's a very nice balance of probabilistic reasoning, and empirical relevance. It is a 'meta'-algorithm that can be adapted and used for multiple search models and basically any oracle out there, which increases it's impact.

Weaknesses: - In most of the paper the problem is described as maximizing the probability of finding x whose y belongs to set S. However, in table 1, it seems like we are mostly back to finding the maximum value possible. I appreciate that these two objectives are very closely related, but they are not the same. I.e. it seems like y_t is set post-hoc. It would be interesting to report the expected probability of belonging to S, when say y_t is the 80th percentile in the table as well. - The value of formulating (or discussing) the algorithm as a game theoretic one is not clear to me, and in fact, raised a concern. Nash's theorem suggests that an equilibrium must exist. However, it is also known that computing Nash equilibria is PPAD-hard, even for two player games (see Daskalakis et al 2008 and extensions). It's therefore clear that the procedure proposed in the paper does not come with good convergence guarantees. This seems to be empirically fine. -It is unclear how well the approach can be extended to implicit generative models. CbAS and DbAS were originally proposed in the sequence design setting, and the fact that the authors opted to use it in the continuous domain may suggest that the method is not yet ready for discrete domain/implicit generative models. The authors mention this as future work. It's helpful to know if the authors has found AF more difficult to use in such setting, or it's simply hasn't been tried.

Correctness: I found no errors in the claims or methodology.

Clarity: The paper is generally well written. While informative, I do find figure 1 a bit overloaded with information to the point of making it impossible to parse without reading the entire legend and associated text. My feeling is that panels e & f (on the search model) can go into the supplement to open room, and showing iterations 0,50,100 is sufficient. Then you can separated the AF and non-AF versions on two clean rows. Another suggestion is to use a different color for the weights of the points in figure 1, as the grey also matched the ground truth and the background, and originally this resulted in confusion. Code is provided, is clean and readable. Thank you!

Relation to Prior Work: Yes. However, I think the following papers as well as a brief discussion of their relevance is suggested. All of them do apply provisions for managing out of distribution pathologies. Kumar and Levine Model inversion networks for model-based optimization. arxiv 2019 Angermueller et al Model-based reinforcement learning for biological sequence design, ICLR 2019 Linder et al Deep exploration networks for rapid engineering of functional dna sequences. bioarxiv 2019 and/or A Generative Neural Network for Maximizing Fitness and Diversity of Synthetic DNA and Protein Sequences Cell systems 2020

Reproducibility: Yes

Additional Feedback: I have read the author responses and believe that the paper is good and meets the criteria for acceptance. I will not change my numerical score as I think per discussion with other reviewers, some empirical concerns remain and a single empirical experiment makes the paper's potential less convincing, especially in discrete design problems where a lot of the motivation for the paper is realized. I strongly believe this paper should be accepted.


Review 3

Summary and Contributions: ===== Please see updates after author response in the "Additional Feedback" area. ===== In this paper, the authors approach the difficult problem in model-based design of drift away from the samples on which the oracle was trained. To overcome this issue, the authors propose to dynamically retrain the oracle through sample reweighting.

Strengths: This work is a great recognition and extremely clear formulation of the problem of a static oracle. In addition, this paper proposes a particular suggested mechanism for retraining the oracle by sample reweighting. This is a novel idea which offers many ways to be extended and can be applied to the rapidly-developing field of model-based design, which is highly relevant to the NeurIPS readership. In addition, the authors provide a very clear description of this method, including pseudocode.

Weaknesses: To me, there are 3 main weaknesses with this work as currently presented: (1) the problem definition, (2) the method limitations, and (3) the empirical results. (1) In the problem formulation, the authors suggest that we should not be allowed to acquire any new samples to retrain the oracle, which disallows any active learning approaches. In practice, which applications does this formulation allow which active learning does not? It would be very nice to know how much we are giving up by disallowing even a single additional query. In addition, the authors formulate the problem as "querying the oracle in regions of the input space which are not well-represented by the oracle training data". However, while this may conjure motivations of extrapolation (changing the input domain), the authors tend to test interpolation (querying within the input domain of training data, but in regions of few samples). Importance sampling will work much better for interpolation rather than extrapolation (and the empirical results seem to be focused on interpolation), so it would be helpful to motivate this vision. (2) The second weakness that I see is regarding the method of importance sampling. The variance of the sample weights explodes exponentially with dimensionality, which will reduce the effectiveness of this approach. The authors seem to recognize this issue, and suggest methods of monitoring the variance. However, monitoring the variance does not reduce the variance. As a result, the authors suggest to use methods which restrict the explored region to a trusted region; this seems in conflict to the goal of exploring beyond the training space. Also, in Eq. 17, when P_0(y \in S) is small, the upper bound on the variance will become vacuous. Unfortunately, this seems to be precisely the situation in which we are most interested in autofocusing. (3) Finally, this paper has limited empirical results. The main simulation study (for which the authors conduct parameter sweeps in the Appendix) is on a 1-dimensional problem. As a major concern is the variance of the sample weights, which has trouble in higher dimensions, it would be very nice if this experiment also swept over dimensionality. In section 5.2, it is not exactly clear what the experiment is testing. The authors select training points which were in the bottom 80th percentile of ground-truth expectations; it would be good to give some measurement of the distance over which the autofocusing must act. Finally, allowing a model to be dynamically retrained could be similar to expanding the model's representational capacity; it would be interesting to see an experiment comparing the results of autofocusing against results from using a larger oracle model. I expect that the autofocusing would substantially outperform the larger oracle model, so this could be an experiment to help argue in favor of autofocusing.

Correctness: Yes, the claims and methods are correct.

Clarity: Yes, this paper is very well written. Clarity is one of the strengths of this submission. Minor points relating to clarity: - Lines 42--49 are a little hard to follow and could be reworded. - p_{\beta} seems to be used interchangeably with P_{\beta} (e.g. line 68 vs line 78) - The aside in lines 82--86 regarding controlling the variance of the Monte Carlo does not seem to add to the discussion of the EDA problem, and could distract a reader from the main point in lines 87-88 that p_{\beta} being fixed constant is a limitation. - The fraction 1/n in Eq. 6 is not necessary given the argmax.

Relation to Prior Work: The authors discuss sufficient related work on model-based design. It seems that the idea of "autofocusing" has connections to the broader ideas of tailoring models to specific environments, such as in meta-learning or contextual learning. Connections to these fields may help readers in understanding why the solutions proposed in those related fields do not solve the problem in this domain.

Reproducibility: No

Additional Feedback: If the authors can sufficiently demonstrate that either (1) this method of sample re-weighting can effectively work even for high-dimensional problems and have some level of extrapolation, or (2) the idea of oracle autofocusing is so useful, novel, and extendable, I could be convinced to increase my score for this submission. As it currently stands, I believe this paper has a very strong problem formulation but I am unconvinced by the proposed solution. ======== Updates after author response and reviewer discussion ======== I am very happy the authors provided an example of using autofocusing in higher-dimensional spaces (60 features). I hope that the authors may find a way to include this experiment in the supplement, with a reference in the main text, so that other readers will be convinced of the applicability of autofocusing to higher dimensions. It would also be great if the simulation experiment which was studied extensively had results for a variety of dimensionalities, but that sort of investigation can wait for another paper. I am thankful that the authors agree that if P_0(y \in S) is small, then the variance will explode and autofocusing will be of limited utility. It seems that there is some "sweet spot" for autofocusing, where the training data must be moderately informative of all areas of interest, but not so informative that the fixed oracle is most useful. I hope that the authors could find a way to discuss this open problem head-on in the camera-ready copy, as I believe this discussion could help readers understand the exact problem that is solved by autofocusing. In discussion with other reviewers, I've been convinced that the "fixed-data" domain is indeed very useful. If space permits, I'd urge the authors to rephrase their discussion of the reasons why the fixed-data domain is necessary. As currently written, "Even if one performs iterative rounds of data acquisition, at some point, the acquisition phase concludes" is not strictly true given that there are many systems which have never-ending acquisition phases. Instead, it may be useful to discuss situations in which the time lag for acquisition is too long, or the data acquisition must be performed in extremely large batches, to preclude any active learning approaches. Overall, I'm now satisfied that autofocusing is a clear framework to move beyond fixed oracles, and I'm also happy that this specific formulation is a great first step. I look forward to future works which probe the empirical utility of this autofocusing method and future development of variance-reduction methods to produce better autofocusing. Overall score: 5 (old) --> 7 (new)

[Author Response · NeurIPS 2020]

Thank-you for the thoughtful and constructive feedback on our manuscript. Below, citation numbers refer to the main
text and new references are made in-line.

We have now run experiments showing that autofocus can still be beneficial in *higher dimensional design spaces*.
For the superconductor design experiment, originally conducted on the 10 most informative features, we have run
the same experiment on 60 features (leaving out the 21 least informative features due to colinearity issues). As a
point of comparison, many Bayesian optimization methods are considered practically useful up to around ten or so
dimensions (Wang, Zoghi, *et al.*, IJCAI 2013). Presently, 60 is already a stretch goal for many problems. We find that
autofocus maintains its statistically significant gains over non-autofocusing for the trust region EDA, CbAS (Table A1)
at dimensionality of 60. We have not yet examined the other methods, but this result alone already demonstrates that
there is no inherent problem to using autofocus in higher dimensions. We will pursue more systematically investigating
the effect of dimensionality, across all methods.

Table A1: As Table 1 in the main text, but with 60 dimensions. (*) means $p$-value $< 0.05$, (**) means $p < 0.01$

| | Median | Max | $\rho$ | RMS |
|---|---|---|---|---|
| | | **CbAS** | | |
| Original | 44.7 | 96.0 | -0.12 | 26.4 |
| Autofocused | 52.0 | 103.4 | 0.12 | 21.6 |
| Mean Diff. | **7.2**** | **7.3**** | **0.23**** | **-4.7**** |

If one defines *extrapolation* as going into regions of the input space where the property (*e.g.*, $\mathbb{E}[y \mid \mathbf{x}]$) takes on very
different values from the training labels, then indeed, this does occur in our experiments. In particular, all autofocused
methods except Random Search always produced *top candidates with ground-truth expectations greater than the*
*maximum training label*. We have now computed the probabilities that the methods produce top candidates with
ground-truth expectations that surpass the maximum training label (a "probability of improvement"). For example,
non-autofocused CbAS yields a probability of improvement that is $28$ times higher than that of a random control (*i.e.*,
drawing the same number of samples from the training distribution) ($p < 0.01$, Mann-Whitney $U$-test). Autofocused
CbAS increases this to $65$ times ($p < 0.01$). As for whether we are interpolating within the training data, we are not in
the sense that the effective sample size (ESS) reported in the original submission shows that the search model deviated
quite considerably from the training distribution (Fig. S3), as the ESS is the inverse of the exponentiated Renyí-2
divergence. Similarly, in additional experiments, we now find that the KL divergence between the training and test
distributions goes up to $150$ for DbAS and up to $10$ for CbAS. All that being said, the goal of AF is a subtle one. We
do not claim that AF *necessarily* makes a design method move further beyond the training data; instead, AF helps
us make the best use of the oracle for wherever a design method does move, regardless of whether that movement
is interpolation or extrapolation (also see "Oracle bias-variance trade-off" paragraph on page 4). From that point of
view, using a method with a trust region complements, rather than contradicts, the goal of AF, although a more formal
examination of how trust-region methods like CbAS can still enable extrapolation would be useful.

With respect to "reporting the expected probability of belonging to $S$, when say $y_\tau$ is the 80th percentile", if we
understand correctly, this is tantamount to asking about extrapolation, and hence addressed in the previous paragraph.
Note too that for maximization design problems, the desired $S$ will be the set of values $y$ such that $y \geq y_{\max}$ (where
$y_{\max} \equiv \max_{\mathbf{x}} \mathbb{E}_{p(y|\mathbf{x})}[y]$).

With respect to *discrete design spaces*, in preliminary experiments on the protein design problem in (9), autofocused
CbAS compared to CbAS yielded candidates with a maximum ground truth of $3.39$ compared to $3.37$ ($p = 0.03$) and a
Spearman's $\rho$ of $0.83$ instead of $0.70$ ($p = 0.01$). This topic requires more comprehensive experimentation which we
continue to do.

The practical importance of the *fixed-data setting* has been demonstrated by recent protein engineering and materials
design work, in which a single labeled dataset is collected, and a regression model is trained and used to guide the
design of novel proteins without additional data (5, 6). Moreover, a prominent protein engineer has told us the fixed-data
setting is what they are currently working on (Frances Arnold, personal communication). Finally, no matter how many
rounds of data are acquired, at some point, the data are fixed and our current problem formulation applies.

With respect to the comment about non-negligible $P_0(y \in S)$, indeed, if this is too small, then the variance explodes. In
effect, this says that the starting (training) data must have at least a hint of the property we care about, just as is required
for successful directed evolution for protein design (3). We will add further clarification in the manuscript.

We will incorporate feedback on clarity of Fig 1, and also on our related work section, in particular linking to
out-of-distribution generalization, small-area estimation, model misspecification, model inversion networks, and the
Angermueller paper *etc*. As for formulating autofocus as a game, this was simply done to give a precise name to
the problem formulation that emerged from our derivation. And indeed, Nash equilibria are hard, but as you note,
empirically, we still find benefit from our approach. Finally, we apologize if some points were not addressed, but in
light of time and space constraints we had to prioritize. We will of course address all points raised.

[Meta-Review · NeurIPS 2020]

A thoughtful and innovative paper that describes an oracle-based method for data-driven design. The referees praise the clarity of the presentation as well as the theoretical sound presentation of the model. Overall, a strong neurips contribution.